# Processing and quality control of FY-3C/GNOS data used in numerical weather prediction applications

Mi Liao[1], **Sean Healy[2],** Peng Zhang[1]

[1] National Satellite Meteorological Center, Beijing, China

[2] European Centre for Medium-Range Weather Forecasts, Reading, UK

Correspondence to:

Sean Healy (sean.healy@ecmwf.int), Peng Zhang (zhangp@cma.gov.cn)

**Abstract**

The Chinese radio occultation sounder GNOS (Global Navigation Occultation Sounder) is on the FY-3C satellite, which was launched on September 23, 2013. Currently, GNOS data is transmitted via the Global Telecommunications System (GTS) providing $450 - 500$ profiles per day for numerical weather prediction applications. This paper describes the processing for the GNOS profiles with large biases, related to L2 signal degradation. A new extrapolation procedure in bending angle space corrects the L2 bending angles, using a thin ionosphere model, and the fitting relationship between L1 and L2. We apply the approach to improve the L2 extrapolation of GNOS. The new method can effectively eliminate about 90% of the large departures. In addition to the procedure for the L2 degradation, this paper also describes our quality control (QC) for FY-3C/GNOS. A noise estimate for the new L2 extrapolation can be used as a QC parameter to evaluate the performance of the extrapolation. A statistical comparison between GNOS bending angles and short-range ECMWF (European Centre for Medium-Range Weather Forecasts) forecast bending angles demonstrates that GNOS performs almost as well as GRAS, especially in the core region from around 10 to 35 km. The GNOS data with the new L2 extrapolation is suitable for assimilation into numerical weather prediction systems.

# 1    Introduction

GNOS is the first radio occultation (RO)sounder on the Fengyun series of Chinese polar orbiting meteorological satellites. It is also the first multi-GNSS (Global Navigation Satellite System) RO receiver in orbit that can perform RO measurements from both GPS (Global Positioning System) and Chinese BDS (BeiDou Positioning System) signals. GNOS is manufactured by National Space Science Center (NSSC) of Chinese Academy Science (CAS), and is operated by the National Satellite Meteorological Center (NSMC) of the China Meteorological Administration (CMA). GNOS is also mounted on FY-3D (which was launched on November 2017) and it will be on all the subsequent Chinese Fengyun satellites. The FY-3 series is expected to provide GNOS RO measurements continuously at least until 2030 (Yang et al., 2012), so this is a potentially important source of data for numerical weather prediction (NWP) and climate reanalysis applications.

As a multi-GNSS receiver, GNOS has the ability of tracking up to eight GPS satellites and four BDS satellites for precise orbit determination (POD). In addition, it has velocity and anti-velocity antennas for simultaneously tracking at most six and four occultations from GPS and BDS, respectively. Because of the presence of two antennas in opposite directions, both the rising and setting occultations can be retrieved. More instrumental details are given in the Table 1, and in Bai et al. (2014). Currently, FY-3C GNOS GPS measurements can produce about 500 GPS-RO profiles per day for operational use in NWP systems, while GNOS from BDS signals are not yet operational, and produce only about 200 profiles because of fewer reference satellites.

As with the pre-existing GPS-RO sounders, such as the GPS/Met (Global Positioning System/Meteorology) experiment (Ware et al., 1996), the COSMIC (Constellation Observing System for Meteorology, Ionosphere, and Climate; Anthes et al., 2008), and the European Metop/GRAS (GNSS Receiver for Atmospheric Sounding) mission (Von Engeln et al., 2009), the raw observations from GNOS consist of phase and signal to noise ratio (SNR) measurements. In addition, auxiliary information provided by the International GNSS Service (IGS), such as the GPS precise orbits, clock files,

Earth orientation parameters, and the coordinates and measurements of the ground stations, are also needed. The IGS ultra rapid orbit products, with an approximate accuracy of 10 cm in orbit, are chosen for near-real-time operational use. The Low Earth Orbit (LEO) precise orbit determination (POD) can be estimated by integrating the equations of celestial motion (Beutler, 2005) using the Bernese software Version 5.0 (Dach et al., 2007). The single difference technique is applied to obtain the excess phase as a function of time in an Earth-centred inertial reference frame. The Radio Occultation Processing Package (ROPP) software (Version 6.0), developed by the EUMETSAT ROM SAF (Radio Occultation Meteorology Satellite Application Facility), is used to determine different atmospheric parameters (Culverwell et al., 2015). One-dimensional variational (1-D-Var) analysis, using background information from a T639L60 global forecast model, is used to retrieve temperature and humidity profiles. The T639L60 is a global medium-range weather forecast system of China, which became operational at CMA in 2009. However, since early 2017, some changes have been implemented in the operational stream. We obtain the auxiliary files through an ftp server in near real time provided by EUMETSAT GSN service, improving the timeliness to within three hours. In addition, the POD software was replaced by the PANDA (Positioning And Navigation Data Analyst), which is developed originally by the Wuhan university of China (Shi et al., 2008).

It is known that GPS signal SNR falls with decreasing altitudes, and especially for the L2 frequency. Montenbruck (2003) and Bergeton (2005) tried to use high quality single frequency to process atmospheric radio occultations without the degraded L2 signal, but have limitations in the condition of high ionspheric oscillations. Dual-frequency for atmosphere radio occultation is still essential. Gorbonov developed an indicator (2005) to estimate the quality of L2 signal in the low atmosphere, and use it to judge where needs to linearly extrapolate the difference of L1 and L2 signal. Z.Zeng (2016) investigates the optimal height for the extrapolation of L1-L2 by modelling the ionospheric bending angle using an approximate

expression. These methods are successfully applied for CHAMP,COSMIC, Metop and other missions. However, the degradation of the GNOS L2 had a large impact on the retrieval quality when the measurements were processed with ROPP. ROPP includes a pre-processing step in order to correct degraded L2 data. The approach is based on Gorbunov et al (2005, 2006). The old approach in ROPP requires the L2 penetrating down into 20km at least. It is hard for GNOS to get the entire L2 signal down into 20km. The reason for GNOS losing L2 signal tracking is that GNOS has a lower SNR compared to other missions. Additionally, the GNOS antenna is smaller and not well located on the satellite. Consequently, we have to use additional cables, which results in a larger decrease of SNR than expected. Therefore, we developed and tested a new L2 bending angle extrapolation method for GNOS data, and implemented it in ROPP.

In this paper, we will describe the new processing of GNOS data that reduces the large stratospheric biases in bending angle and refractivity, and present a quality control scheme for FY3C/GNOS. These results will be useful for understanding the statistical error characteristics and quality control of the GNOS data, and more generally the extrapolation approach may useful for other missions where one signal is lost early.

## 2    Large biases in the original GNOS processing

The ROPP software (Culverwell et al., 2015) is used to retrieve atmospheric parameters, such as bending angle, refractivity, dry temperature, temperature and humidity, from GNOS excess phase measurements. The geometrical optics approach (e.g., Kursinski et al., 1997) is used to process the L1 and L2 phase delays to bending angle space above 25 km, and the Canonical Transform 2 (CT2)(Gorbunov and Lauritsen, 2004) technique is used for both L1 and L2 signals below 25 km. The combined statistical optimisation ionospheric correction method (Gorbunov 2002) produces "optimised" bending angles that are subsequently used in an Abel transform to produce refractivity profiles. We note that most NWP centres assimilate either

1    bending angle or refractivity profiles.

In the preliminary assessments for the FY-3C/GNOS GPS RO refractivity retrievals

against NWP with the original ROPP processing system, it was found that the most

obvious and prominent quality issue was the large departure biases, in the vertical

range of 5-30 km, peaking at around 20km (Figure 1). The percentage of profiles

affected was about 13~15%. This bias problem is not seen with other RO missions,

and it was found to be related to GNOS GPS L2 signal tracking problems, and the

subsequent extrapolation of the L2 signal. It was found that most of the bad GNOS

cases are rising occultations.To improve the tracking in the lower troposphere and the

quality of rising occultations, open loop tracking is implemented for GNOS GPS L1

signal, but not for L2 (Ao et al., 2009). In general, SNR falls under the complicated

atmospheric conditions in troposphere because of atmospheric defocusing. The GPS

L2 signal is modulated by a pseudo-random precision ranging code (P code) for the

purpose of anti-spoofing. Although GPS L2 can be demodulated using the

semi-codeless method, it will be at the expense of SNR and precision (Kursinski et al.,

1997). Therefore, the performance of L2 signal tracking is not as good as that of L1,

especially for the rising occultations. Figure 2 shows the lowest Straight Line Tangent

Altitude (SLTA) percentages of L1 and L2 signals, for both the rising and the setting

occultations. It shows that the lowest tracking height of L1 C/A of both the rising or

setting measurements are reasonable (Sokolovskiy,2001), with more than 98.5%

profiles with a below zero SLTA. However, for the L2P, only 70% of the rising

measurements reach below 20km. There are 24.8% of rising profiles stopping in the

range of 20 ~70km, and 5.2% stopping above 70km, meaning effectively they contain

no valid measurements. In contrast, 89.9% of setting occultations can get below 20km,

which is better than the rising, but about 10% stop above that height. Those profiles

that have bad L2 signal observations significantly affect the retrievals when using

ROPP software to process the GNOS data.

Figure 3 shows an example of GNOS performance in terms of excess phase, SNR,

and bending angle for two bad cases where the L2 stops early. In these two cases,

there are no valid L2 excess phase observations below 25km or 30km SLTA, respectively. However, there are L2 bending angles, extending to the near surface because of extrapolation within ROPP.Figure 4 is the same as Figure 3 but for two good cases where the L2 measurements get to 20km SLTA. Compared with the bad cases, the retrieved bending angles of L1, L2 and LC span a similar vertical interval , and show good agreement even at the lower part of the profiles.

ROPP includes a pre-processing step designed to correct degraded L2 data. The approach is based on Gorbunov et al (2005, 2006), and it is used successfully for other GPS-RO missions. Briefly, smoothed L1 and L2 bending angle and impact parameters are computed. An impact height, "PC", above which the L2 data is considered reliable, is estimated using an empirical "badness score". The empirical badness score at time t, is defined as,

$$Q(t) = \left( \frac{abs(\overline{p_1(t)} - \overline{p_2(t)})}{\Delta p_a} + \frac{\delta p_2(t)}{\Delta p_b} \right)^2 \qquad (2.1)$$

where $\delta p_2$ is a measure of the width of the L2 spectrum, $\overline{p_1(t)}$ and $\overline{p_2(t)}$ are the L1 and L2 impact parameters, respectively, computed from smoothed timeseries, $\Delta p_a$=200 m and $\Delta p_b$=150 m (See also, Eq. 11 Gorbunov et al, 2006 for a slightly modified form). The largest $Q(t)$ value in the impact height interval between 15 km to 50 km is stored as the badness score for the occultation, potentially for quality control purposes.

The mean L1 and L2 bending angle and impact parameters are then computed in a 2 km impact parameter interval directly above PC. Simulated L2 bending angles and impact parameters are computed by adding the mean (L2-L1) differences to both the L1 bending angle and impact parameter values, using the data in the 2 km interval. Simulated L2 and L1 phase values are then computed from these bending angles. Corrected L2 excess phase values are computed by merging the observed L2 phase

above PC, with the simulated values below PC, using a smooth transition over 2 km, centered on PC. The corrected L2 phase values are subsequently used in the wave optics processing of the L2 signals.

A specific difficulty with the GNOS processing is related to determining the impact height PC, used for both the computation of the mean L1 and L2 differences, and defining the transition between observed and modelled L2 phase values. Although the "badness score" is used to determine PC, PC also has a maximum value (20 km). This is defined as the wave optics processing height (25 km) minus a 5 km "safety border". Therefore, the mean bending angles and impact parameters used in the L2-L1 correction can only be computed in a 2 km interval up to a maximum impact height of 22 km. Unfortunately, this is not high enough for GNOS L2 signals, with the result that the mean L2-L1 bending angle and impact parameters computed in the 2 km interval above PC are corrupted, prior to the extrapolation.

## 3   New L2 extrapolation

As mentioned in the Section 2, some form of extrapolation of the observed L2 signal is required before it can be combined with the L1 signal, in order to remove the ionospheric contribution to the bending. However, the current L2 extrapolation implemented in ROPP leads to large bending angle and refractivity departures when processing GNOS RO data. Therefore, an alternative L2 extrapolation method has been implemented in the ROPP to solve the GNOS problem. The new approach is based on (unpublished) work by Culverwell and Healy (2015), who modelled the bending angles produced by a Chapman layer model ionosphere, and established some basic theory for the relationship between fitting L1 and L2. A key underlying assumption in the L2 extrapolation approach is that the total bending angle can be written as a linear combination of the neutral bending plus a frequency dependent ionospheric bending term. Therefore, we assume that subtracting the L1 bending

angle from the L2 value at a common impact parameter, removes the neutral bending

contribution. This is a common assumption, and it is also made in the standard

ionospheric methods used in GPS-RO (Vorob'ev and Krasil'nikova, 1994).

The extrapolation method adopted here is based on a "thin" ionospheric shell

model, where the ionosphere approaches a Delta function, at a specified height (See

section 3.1, Culverwell and Healy, 2015). This ionospheric model is crude, and it

clearly would not be appropriate  if we were attempting to retrieve ionospheric

information. However, in the context of GNOS processing, we are mainly interested

in modelling the impact of the ionosphere on bending angles with a tangent height

well below the ionosphere, typically in the 25-60 km vertical interval. The neutral free

L2-L1 bending angle differences in this interval vary slowly with height (impact

parameter) (e.g., see Figures 2 and 3, Zeng et al, 2016). For example, adding a

sporadic E layer near 100 km would not change the shape of the L2-L1 difference

curve below 60 km significantly. Conversely, we cannot retrieve an E-Layer

information from the L2-L1 differences below 60 km.

Thus, for a vertically localized region of refractivity, sited well above tangent points

of interest, the ionospheric contribution to the bending angle, $\alpha$, at frequency $f$ can be

simply expressed by (Eq. 2.6, Culverwell and Healy, 2015):

$\alpha(a) = 2a\frac{k_4}{f^2}\int_a^\infty \frac{xn_e(x)}{(x^2-a^2)^{\frac{3}{2}}}dx$     (3.1)

where $x = nr$, is product of the refractive index, $n$, and radius value $r$, $a$ is the

impact parameter, $k_4 = \frac{e^2}{8\pi^2 m_e \varepsilon_0} = 40.3 m^3 s^{-2}$, and $n_e$ is the electron number density.

Commonly, the electron number density can be expressed in terms of the vertically

integrated total electron content, TEC, which is defined as $TEC = \int n_e dr$. The

equation above can be simplified by assuming a very narrow ionospheric shell and

written as (Eq. 3.2, Culverwell and Healy, 2015):

$\alpha(a) = 2a\frac{k_4}{f^2}TEC\frac{r_0}{(r_0^2-a^2)^{\frac{3}{2}}}$     $(for\ a < r_0)$   (3.2)

$r_0$ is height of the peak electron density, which is assumed to be 300 km above the

surface in this work.

The GPS L1 and L2 frequency bending angle difference is expressed as:

$$\alpha_2(a) - \alpha_1(a) = 2ak_4 TEC(\frac{1}{f_2^2} - \frac{1}{f_1^2})\frac{r_0}{(r_0{}^2 - a^2)^{\frac{3}{2}}} \qquad (3.3)$$

If we define $x_{so} = 2ak_4 TEC(\frac{1}{f_2^2} - \frac{1}{f_1^2})$, then,

$$\alpha_2(a) = \alpha_1(a) + x_{so}\frac{r_0}{(r_0{}^2 - a^2)^{\frac{3}{2}}} \qquad (3.4)$$

In this work we estimate $x_{so}$ from a least-square fit based on observed L1 and L2 bending angle differences produced with geometrical optics, over a 20 km vertical above the lowest valid L2 bending angle value. The maximum height of the vertical interval is limited to be 70 km. In theory, for a spherically symmetric ionosphere, $x_{so}$ should be proportional to the ionospheric TEC, because the L2-L1 differences should be proportional to the TEC. However, we are not trying to retrieve the TEC here, and the quality of the TEC estimates has not been assessed. We simply estimate the parameter $x_{so}$ in order to extrapolate the L2-L1 differences below 25 km using a reasonable, physically plausible curve.

We currently assume the Delta function ionospheric model peaks at 300km above the surface.. Experiments testing the sensitivity of the extrapolated bending angles to changes in the peak height from 250km to 350km, in 10 km increments have been performed. The largest differences between the 250 km and 350 km experiments about 1.0 microradians near the surface (Figure not shown). To put this in some context, the corrected bending angle value at an impact height of 20 km is typically 1600 micro-radians, and the neutral bending grows exponentially towards the surface, with the density scale-height (~7 km). Therefore, the sensitivity to the assumed peak height is low.

Two bad profiles, where the L2 signal stops above 20 km SLTA, have been chosen for demonstrating the extrapolation method. Their detailed information is listed in Table 2. Because the ionospheric effect becomes smaller in relative terms with the

decreasing height, the magnitude of the relative L2-L1 bending angle differences gets smaller with height. Seen from the direct comparisons between the new and the old extrapolation results of case 1 (Figure 5 and 6), L2 bending angles are very different from the L1 bending angles before correction. After applying the new extrapolation approach, the L2 bending angles below 20 km are consistent with both L1 and LC bending angles. It is concluded that a more reliable LC bending angle can be obtained by using the new L2 extrapolation approach than the original L2 extrapolation method implemented in ROPP.

Clearly, using the new simple ionospheric model for the L2 extrapolation performs very well for the bad profiles with large biases. It is also useful to demonstrate the new extrapolation method for normal cases. Here the normal profiles are defined as the lowest SLTA reaching below 20 km, and the mean standard deviation to the reanalysis data is within 2% from surface to 35 km. Therefore, two good profiles (Table 3) are selected to test the new extrapolation.

Generally, the new extrapolation method does not degrade the good profiles. In fact, the new method smooths some occultation points, and improves the consistency of L1 and L2, as shown in Figure 7 and 8, for example.

An alternative way to demonstrate the accuracy of the different extrapolation methods is to compare their refractivity retrievals with the forecast model data. One day of data is used to test the new L2 extrapolation method. Figure 9 shows that the new method can effectively eliminate ~90 % of the problematic "branches" with the large percentage refractivity errors often are exceeding 100 %. In this plot, eight profiles still have a large bias after the new extrapolation, because the L2 SLTA stops above 70 km, which is out of the processing range used in the extrapolation (below 70 km). These cases can be removed by using a simple QC.

## 4   Quality control methods

Based on the GPS RO error sources and characteristics, many internal QC methods have proposed in the literature. For example, the COSMIC Data Analysis and Archive

Center (CDAAC) define an altitude, $Z$, below which a low quality of L2 signal has been detected. The maximum difference of Ll and L2 bending angle above Z, and the ionospheric scintillation index analyzed from the amplitude of L1 signal at high altitudes are used in the QC (Kuo et al., 2004). Gorbunov (2002) proposed a QC procedure in terms of the analysis of the amplitude of the RO data transformed by the Canonical Transform (CT) or the Full Spectrum Inversion (FSI) method (Gorbunov and Lauritsen, 2004), which is useful to catch the corrupted data because of phase lock loop failures. Beyerle et al. (2004) also suggested a QC approach to reject the RO observations degraded by ionospheric disturbances based on the phase delay of L1 and L2 signals. Zou et al (2006) use the bi-weight check, removing large departure data from the statistical point of view. More recently, Liu et al (2018) introduced a local spectral width based quality control, which improves the application in lower troposphere. The quality indicator "badness score" in ROPP is successfully applied for CHAMP, COSMIC, METOP and other observations. However, just like the failure of processing GNOS data, the badness score is not adequate for identifying the GNOS data. The reason might be related to the empirical parameters (see formula 2.1). These parameters are formed based on the performances of CHAMP, COSMIC and METOP missions, whose L2 signals are not degraded too much as GNOS. Considering the new L2 extrapolation method and the characteristics of GNOS data, we introduce a new indicator to detect the poor quality profiles based on the noise estimate of the L1 and L2 fit.

## 4.1  Noise estimate of the L1 and L2 fit

As noted earlier, as a result to L2 signal tracking problems, around 15% profiles are degraded with the old processing. After applying the new L2 extrapolation method, most of them can be effectively corrected. As seen from the Eq. 3.4, the key to the correction is how well the retrieved parameter, $x_{so}$, fits the difference of L1 and L2 bending angles in the 20km fitting interval. Currently, 25 km or the minimum L2 SLTA is the lower limit of the fitting interval.

We have introduced a new parameter, $\theta_\alpha$, to test the quality of the least-square

fit in the 20 km interval. It can be expressed as:

$$\theta_\alpha = \sqrt{\dfrac{\sum\left(x_{so}*\dfrac{r_0}{(r_0{}^2-a^2)^{\frac{3}{2}}}-\Delta\alpha(a)\right)^2}{n}} * 10^6 \quad (4.1)$$

where $\Delta\alpha$ is the difference of L1 and L2 bending angles, and the sum is over the $n$

(L2-L1) values in the 20 km fitting interval. The parameter $\theta_\alpha$ is the

root-mean-square of the difference between the fitting model and (L2-L1) values.

Clearly, it provides information about how well we are able to fit the L2-L1 bending

angle differences with the model, in a fitting interval where we trust the data. We

assume that if the fitting model can reproduce the L2-L1 bending angle differences

accurately in the fitting interval, we can then use the retrieved parameter $x_{so}$ to

extrapolate the L2-L1 differences below 25 km, to produce reasonable ionospheric

corrected bending angles used for NWP applications.

A histogram of the $\theta_\alpha$ values has been obtained by accumulating statistics over a

seven day period (Figure 10), and we use this to determine a QC threshold value as 20

microradian. Clearly, the 20 microradian threshold is empirical, but it can be related

tothe assumed bending angle error statistics used in the assimilation of GNSS-RO

data. At ECMWF, the assumed bending angle uncertainty is 1.25 % from around 10

km to ~32 km, and the 3 micro-radians above this height.This translates into around

7.5 microradians at 26 km, increasing to around 20 micro-radians at 20 km. The 20

microradian threshold is designed to screen out cases where the L2-L1 extrapolation

could introduce significant additional errors. In summary, in the operational GNOS

processing, if the value of the $\theta_\alpha$ is greater than 20 microradians, the profiles will be

rejected.

## 4.2   Mean phase delays of L1 and L2

The $\theta_\alpha$ QC parameter does not detect all the poor quality profiles, and we need

additional quality control methods to identify them. We find that it is also necessary

to monitor the performance of GNOS mean L1 and L2 phase delays in the height

interval of 60 to 80 km, because this can also indicate the observational quality of GPS RO data. However, the L1 and L2 SNR values, which are commonly used as a QC indicator, are not found to be useful for identifying the large bias cases of GNOS data. For the rising profiles, the absolute accumulated phase delay should increase with height. Despite reasonable SNR above the height of 60km, in some cases the mean phase delays have small values, leading to problems in the inversions.Figure 11 and Figure 12 show the histograms of the L1 and L2 mean delay phase in rising occultations. They show that there is a clear separation of the mean phase delays. To clarify the quality of the two groups of samples, we identify them as "GOOD" or "BAD" profiles. The criterion for good or bad is that the mean bias relative to the background data is smaller than or greater than 2% at the height interval of 10 to 40km, respectively. Figure 13 and 14 demonstrate the distribution of L1 and L2 mean phase delay. Different colour represents different overlap density, the dark blue is the lowest density and the dark red is the highest one. The colours between them represent increasing density. The "GOOD" samples gather around -8000 meters, while the "BAD" samples accumulate around -100 meters. Therefore, we can identify most of the bad rising occultations, when both L1 and L2 absolute mean phase values are smaller than 150 m. This threshold value is empirical considering the amount of the samples. Unavoidably, a small number of good profiles could be wrongly detected as well and few bad ones could be missed.

**4.3 The statistical performance of the applied QC methods**

After checking a number of QC parameters, we use the following three QC tests:

(1) If the occultation is rising, and the absolutemean phase delays of L1 and L2 are both smaller than 150m, the profile will be identified as "bad";

(2) If the value of $\theta_\alpha$ is greater than 20 microradians, the profile will be identified as "bad";

(3) If the lowest SLTA of L2 is greater than 50 km, the profile will be identified as "bad".

These have been tested with three months of data, as to whether they can identify

the "good" or "bad" large bias cases. The criterion for good or bad is similar to those

mentioned above that the mean bias relative to the background data is smaller than or

greater than 2% at the height interval of 10 to 40km, respectively .

41,928 samples are collected from April 1 to June 30, 2018. There are 38,752

good profiles and 3,176 bad profiles evaluated by background data (e.g. The ECMWF

reanalysis). The QC scheme applied in this paper identifies 37,627 good profiles and

4,301 bad ones. According to statistics, the number of profiles that can be accurately

identified is 36,957, the accuracy rate is 95.4%, the number of missed is 1,795, the

missed rate is 4.6%, 670 are misjudged, and the false positive rate is 1.8%. See Table

4 for clarification. Unavoidably, a small number of good profiles could be wrongly

detected as well and few bad ones could be missed. In general, the performance of

this kind of QC method can effectively identify most of the bad profiles.

**5    Comparison with ECMWF forecast data**

This section demonstrates the performances of the comparison between the

observational GNOS bending angles and the simulated ones using ECMWF

short-range forecast data. GNOS bending angle profiles are those which are carried

out using the new L2 extrapolation and quality controls mentioned in section 3 and

section 4, respectively. The period is from 6[th] July to 2[nd] Aug. 2018. The ECMWF

data used as the background is the state-of-the-art short-range forecast data with 137

vertical levels extending from surface to 0.01 hPa. Using the 2D bending angle

forward operator, ECMWF forecast data can be projected into the bending angle

space at the GNOS locations.

GNOS observations are provided BUFR format for NWP applications, with the

bending angles given on 247 vertical levels from the surface to 60 km. To provide a

context for the comparisons, Metop-A GRAS profiles from the same period are also

selected as a benchmark. Figure 15 displays the mean bias for the GNOS and GRAS

bending angle profiles both separated into rising and setting occultations, showing

that GNOS and GRAS are very consistent with each other above 10 km. Figure 16

shows the standard deviation of the bending angle departures for the GNOS and

GRAS. Their standard deviations are about 1% between 10 – 35 km, increasing to

about 12% at 50 km and more than 15% below 5 km impact height. It is clear that the

GNOS standard deviations are comparable to GRAS in the 10 - 40km interval. The

difference in the 20 to 25 km interval is related to the transition from wave optics to

geometric optics for the GNOS. Generally, the two datasets have similar error

characteristics in terms of both the mean bias and standard deviation over most of the

height interval, but especially in the GPS-RO core range between 10-35 km. The

standard deviations of the GNOS departures below 10 km are smaller than the GRAS

statistics. However, we do not believe that this indicates that GNOS data is superior to

GRAS below 10 km. In general, GRAS measurements tend to penetrate more deeply

in the troposphere, and this will affect the statistical comparison with GNOS.

Furthermore, the difference between the setting and rising GRAS statistics is known

but not fully understood, and it is an area of current investigation. Nevertheless, we

believe that Figures 15-16 provide evidence that the GNOS and GRAS measurements

have similar performance in the "core region" as a result the processing and QC

methods introduced here.

Note that further GNOS occultation departure statistics, including comparisons

with other GPS-RO measurements in bending angle space,    are now routinely

available from the ROM SAF web pages.

See,http://www.romsaf.org/monitoring/matched.php

## 6   Conclusions

This study has focused on three main areas. Firstly, we have developed and tested a

new L2 extrapolation for GNOS GPS-RO profiles. Secondly, we have investigated

QC method for GNOS after applying the new L2 extrapolation. Thirdly, we have

estimated the bending angle departure statistics by comparing GNOS and ECMWF

short-range forecast data. The main results are summarized below.

We have identified and investigated the GNOS GPS-RO cases that fail quality control

with large bending angle departures, after the processing with the ROPP software. These large departures can be attributed to the GPS L2 signal tracking problems for signals that stop above 20 km in terms of SLTA, and the related L2 extrapolation. The percentage of the profiles with large departure is about 13~15%. Therefore, we focused on a better L2 extrapolation for GNOS when the L2 signal stops early. A new L2 extrapolation approach has been implemented in ROPP to mitigate the problem. (These modifications will be available in ROPP 9.1; see http://www.romsaf.org/ropp/) The main procedure is in bending angle space, and it is based on the (unpublished) study of Culverwell and Healy (2015). The new method can effectively remove about 90% of the large departures. The remaining poor cases are mostly due to the L2 being completely missing.

We have studied and established the quality control method suitable for GNOS GPS-RO profiles after correcting the large departures. The new L2 extrapolation $\theta_\alpha$ value can be taken as a QC parameter to evaluate the performance of the extrapolation. It is the root-mean-square of the difference between the fit and observations above the extrapolated height. The 20 microradian threshold is used to judge the good or bad profile after implementing the new L2 extrapolation method. The mean phase delays of L1 and L2 in the tangent height interval of 60 to 80 km are analysed and applied in the QC as well. The lowest SLTA of L2 is also set as a threshold to identify the bad profiles. Using the parameters mentioned above, the QC method can correctly identify 95.4% of the profiles.

Finally, we have assessed the quality of the GNOS bending angles after implementing the new processing and QC by comparing with the background bending angles computed from the operational ECMWF forecasts. GRAS profiles from the same period are selected as a benchmark. The departure statistics for the GNOS and GRAS bending angle profiles in terms of the mean bias and standard deviations are similar at most of the heights, especially in the GPS-RO core region between 10-35 km.

As a result of this work, the GNOS data are now assimilated in operational NWP systems at, for example, the European Centre for Medium-Range Weather Forecasts (ECMWF), Deutscher Wetterdienst (DWD) and the Met Office.

**Acknowledgments**

This work was undertaken as part of a visiting scientist study funded by the Radio Occultation Meteorology Satellite Application Facility (ROM SAF), which is a decentralised processing centre under the European Organisation for the Exploitation of Meteorological Satellites (EUMETSAT).

We have to express our appreciation to Christian Marquardt for his valuable suggestions with respect to the RO processing and QC methods. In addition, we want to thank Ian Culverwell and Chris Burrow for their discussions. Finally, we would like to thank the fund support of National Key R&D Program of China (No.*2018YFB0504900*) and Special Fund for Meteorology Research in the Public Interest (No.*201506074)*.

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

<p>Table 1 Main instrumental parameters for FY-3C/GNOS</p>

| Parameters | FY-3C/GNOS |
|---|---|
| Orbit Height | ~836 km |
| Orbit Type | sun synchronous |
| Inclination | 98.75 ° |
| Spacecraft mass | ~750kg |
| Instrument mass | 7.5kg |
| Constellation | GPS L1 C/A, L2 P |
| | BDS B1I,B2I |
| Channels | GPS:14 BDS:8 |
| Sampling | POD 1Hz |
| | ATM.occ. (closed loop)50Hz |
| | ATM.occ.(open loop) 100 Hz |
| | ION occ. 1Hz |
| Open loop | GPS L1 C/A |
| Clock stability | $1\times10^{-12}$(1secAllan) |
| Pseudo-range precision | ≤30cm |
| Carrier phase precision | ≤2mm |
| Beam width of atmosphere occultation antenna | ≥±30°(azimuth) |

Table 2. Details of the selected bad occultations

| No. | Occ. time (yymmdd.hhmm) | Longitude (deg.) | Latitude (deg.) | Occ. direction | SLTA_L2 (km) |
|---|---|---|---|---|---|
| 1 | 170128.0332 | -99.154 | 25.070 | rising | 21.917 |
| 2 | 170128.0740 | 24.705 | -4.222 | rising | 25.793 |

Table 3. Details of the good profiles

| No. | Occ. time (yymmdd.hhmm) | Longitude (degree) | Latitude (degree) | Occ. direction | SLTA_L2 (km) |
|---|---|---|---|---|---|
| 1 | 20170128.0103 | 149.508 | -38.445 | rising | 4.011 |
| 2 | 20170128.0251 | 70.857 | -51.463 | rising | 12.928 |

Table 4. The $2\times2$ table values

| | | Evaluated by background data | |
|---|---|---|---|
| | | GOOD (38752 profiles) | BAD (3176 profiles) |
| Identified by QC parameters | GOOD (37627 profiles) | 36957 (hits) | 670( misses) |
| | BAD (4301 profiles) | 1795(false identified) | 2506(correct negatives) |

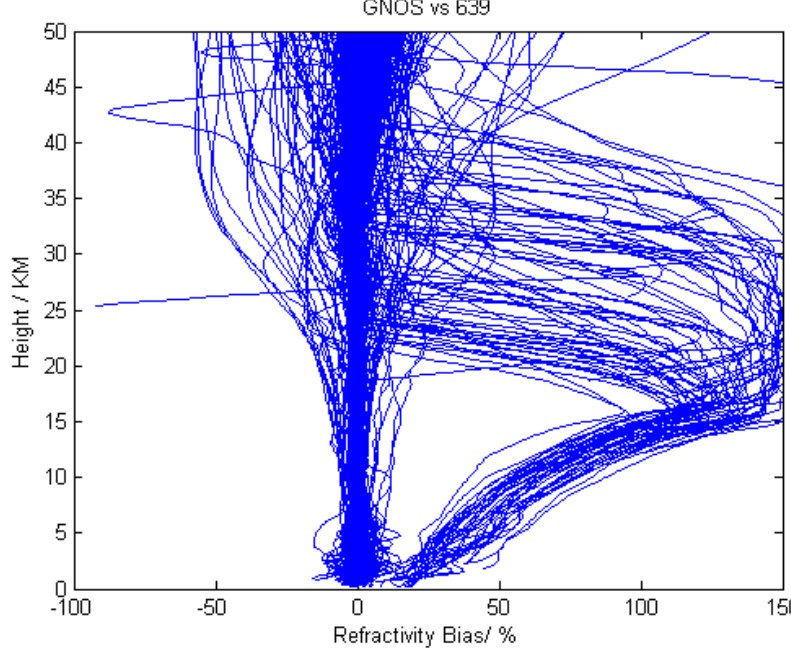

Figure 1. FY-3C/ GNOS GPS refractivity bias compared to T639 (the Chinese
forecast model data), on 28th Jan.2017 with 489 samples.

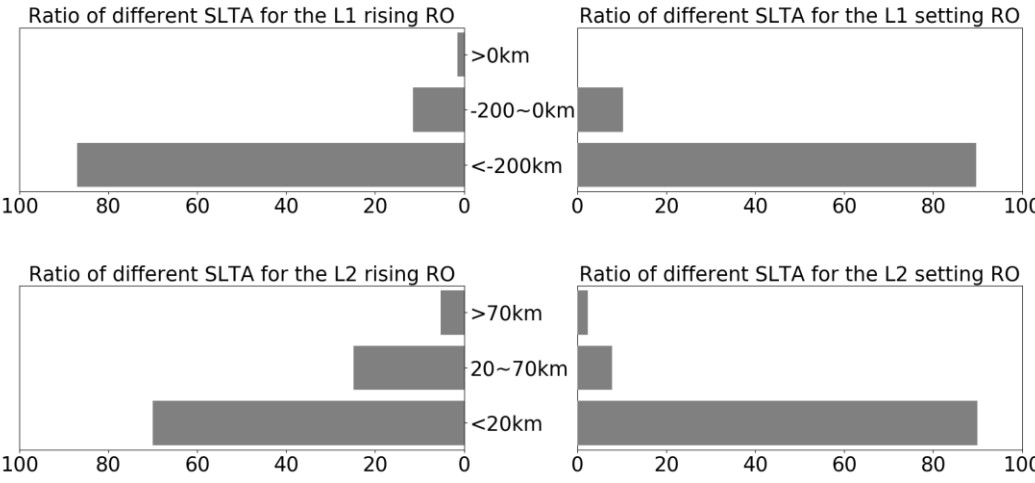

Figure 2. Ratio of different SLTA of the L1 C/A and L2 P for the rising and setting
occultations, statistics result is from 28th Jan to 2nd Feb. 2017.

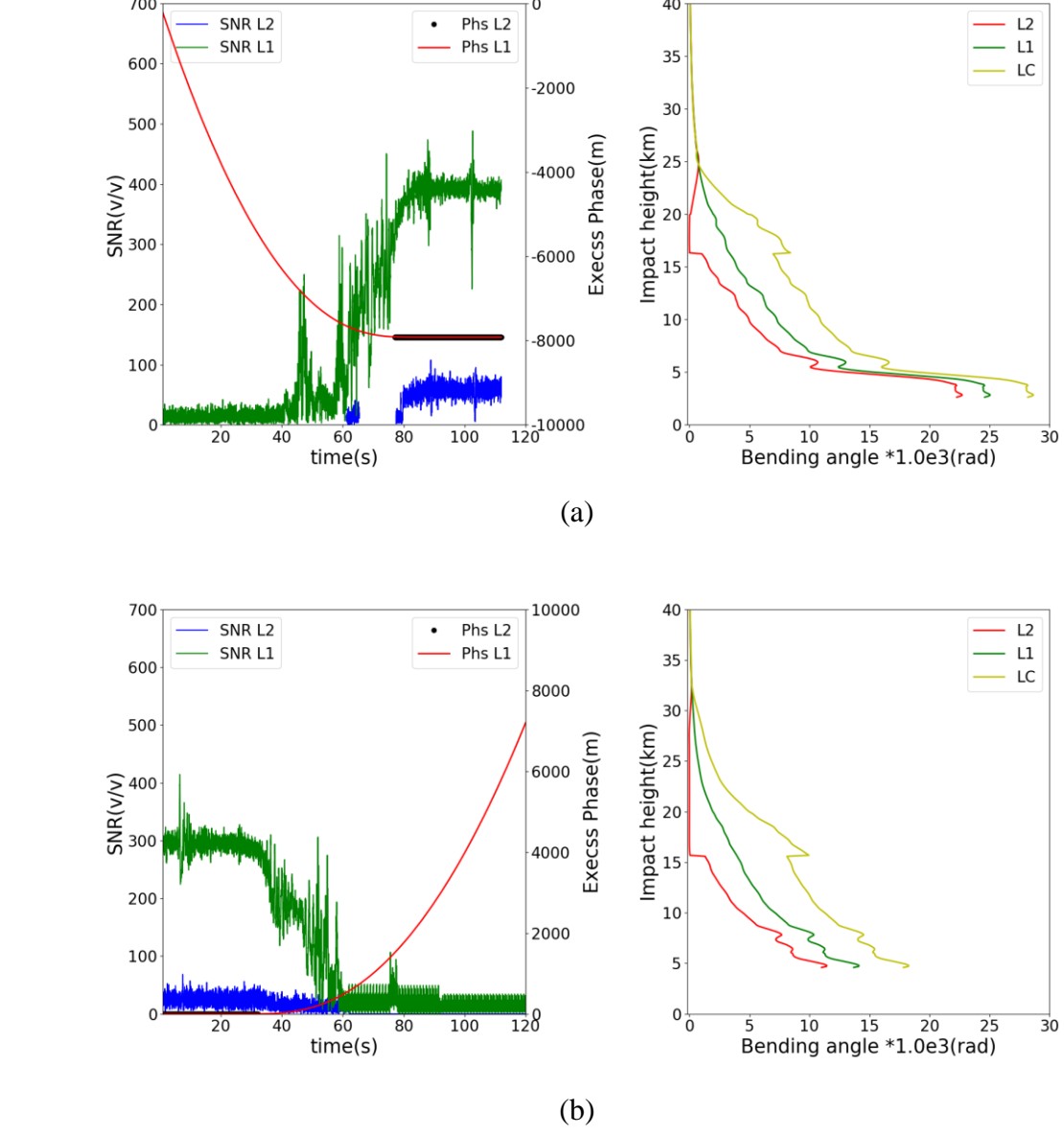

(a)

(b)

Figure 3. Two bad cases (a) A rising profile
(FY3C_GNOSX_GBAL_L1_20170128_0332_AEG15_MS.NC), (b) a setting profile
(FY3C_GNOSX_GBAL_L1_20170128_0850_AEG18_MS.NC). Example L1 (red)
and L2 (black) SNR and excess phase measured data. The resulting L1 bending angle
(green), L2 bending angle (red), and LC bending angle (yellow) profiles as a function
of impact parameter computed using ropp_pp routines.

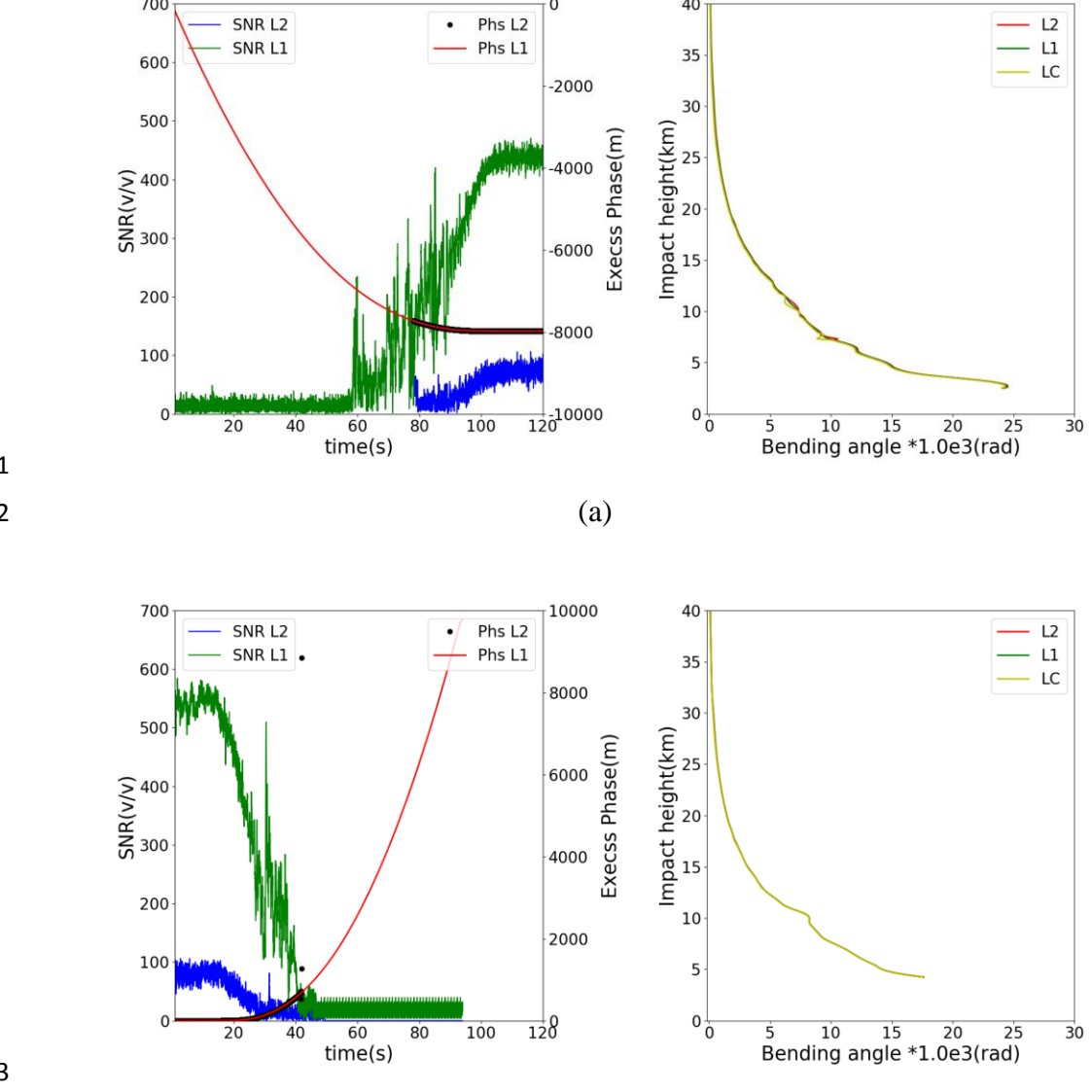

2                                                        (a)

4                                                        (b)

Figure 4. Two good cases (a) A rising profile
(FY3C_GNOSX_GBAL_L1_20170128_1138_AEG27_MS.NC), (b) a setting profile
(FY3C_GNOSX_GBAL_L1_20170128_1648_AEG31_MS.NC). Example L1 (red)
and L2 (black) SNR and excess phase measured data. The resulting L1 bending angle
(green), L2 bending angle (red), and LC bending angle (yellow) profiles as a function
of impact parameter computed using ROPP routines.

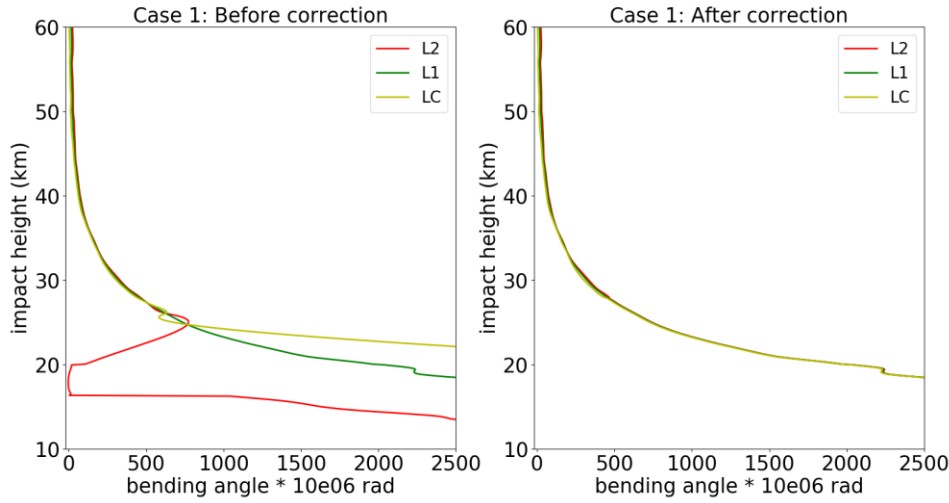

Figure 5. Case1: the bending angle of L2 (red), L1 (green) and LC (yellow) before (right) and after (left) correction.

( FY3C_GNOSX_GBAL_L1_20170128_0332_AEG15_MS.NC)

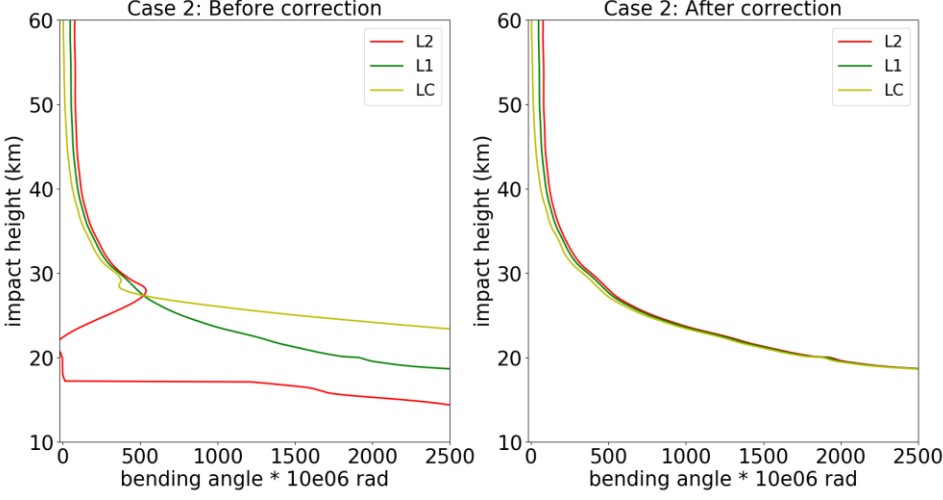

Figure 6. The same as Figure 5 but for Case 2.

( FY3C_GNOSX_GBAL_L1_20170128_0850_AEG18_MS.NC)

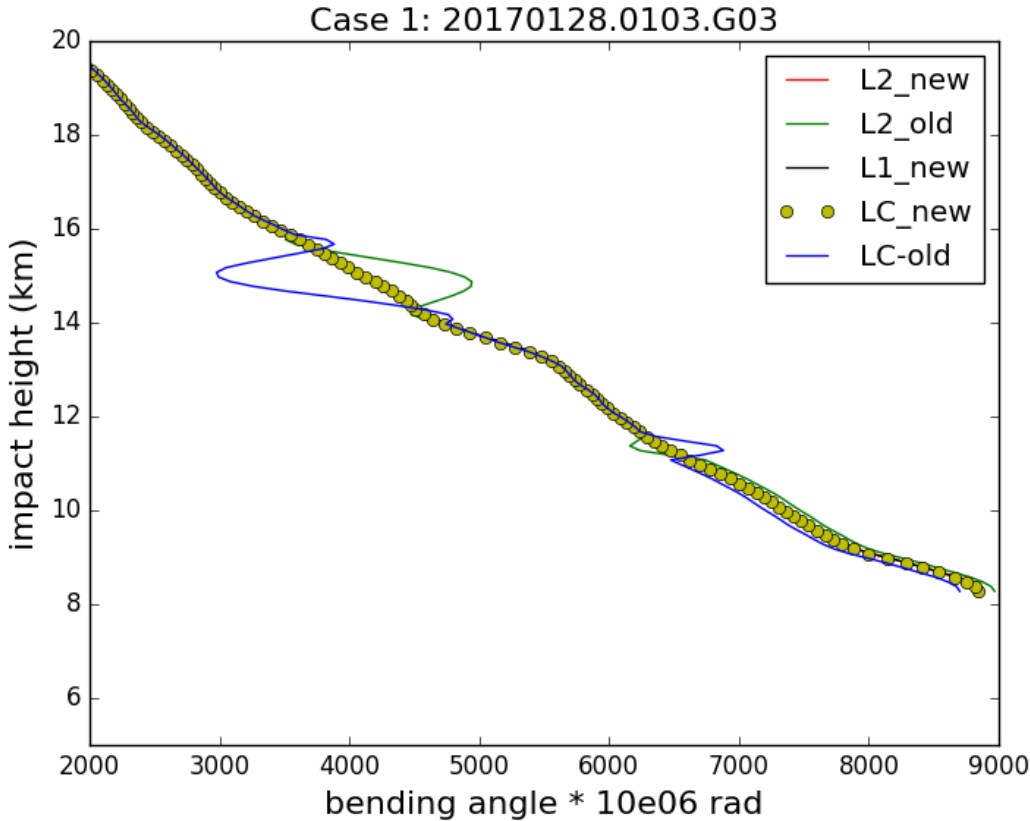

Figure 7. Good Case 1: the bending angle of L2, L1 and LC before and after correction.

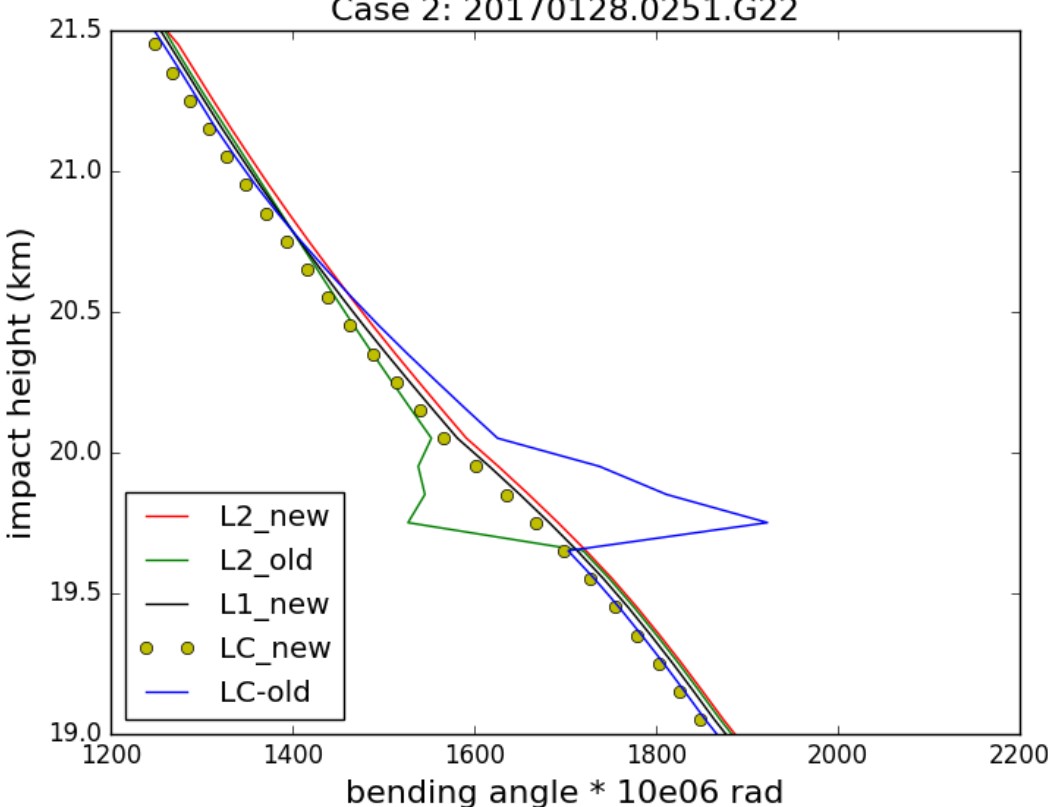

Figure 8. Good Case2: the bending angle of L2, L1 and LC before and after correction.

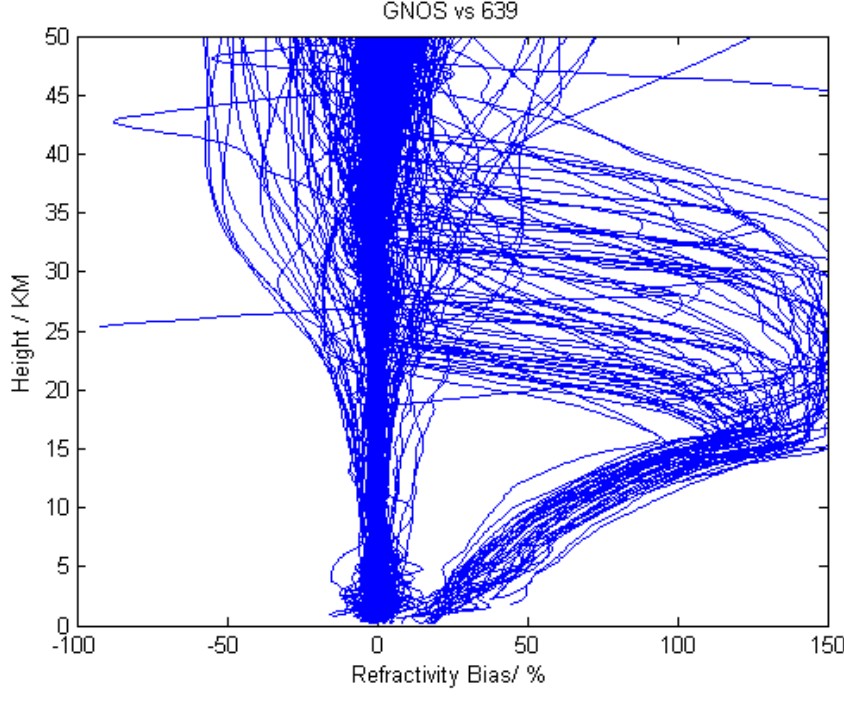

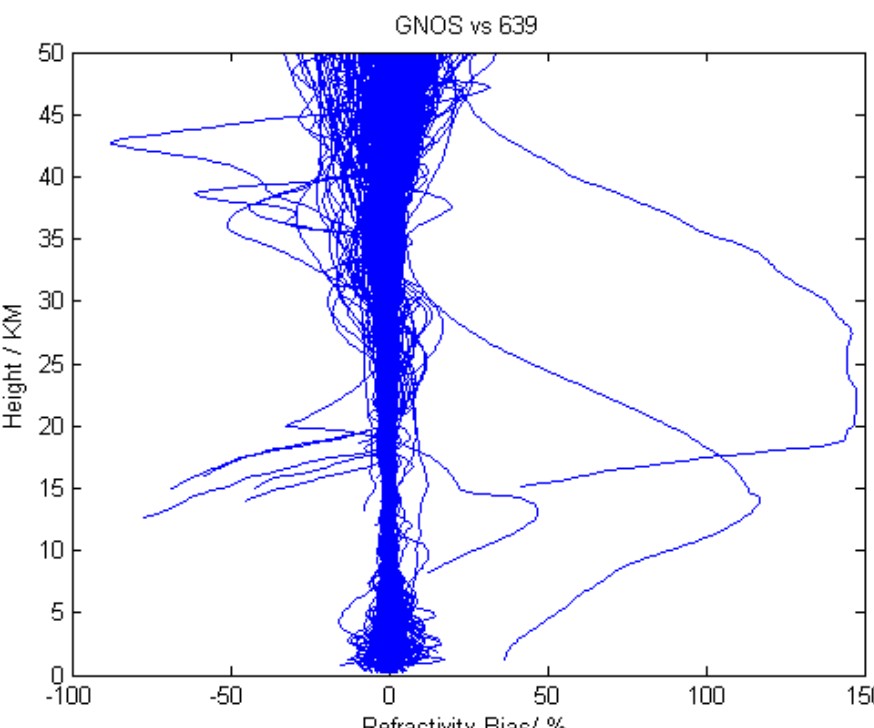

3    Figure 9.   FY-3C/ GNOS GPS refractivity bias compared to T639 (the Chinese

4    forecast model data), on 28[th] Jan.2017 with 489 samples. The upper plot reproduces

5    Figure 1 and is the result of the original GNOS GPS data, and the lower plot isafter

6                    implementing the new L2 extrapolation approach.

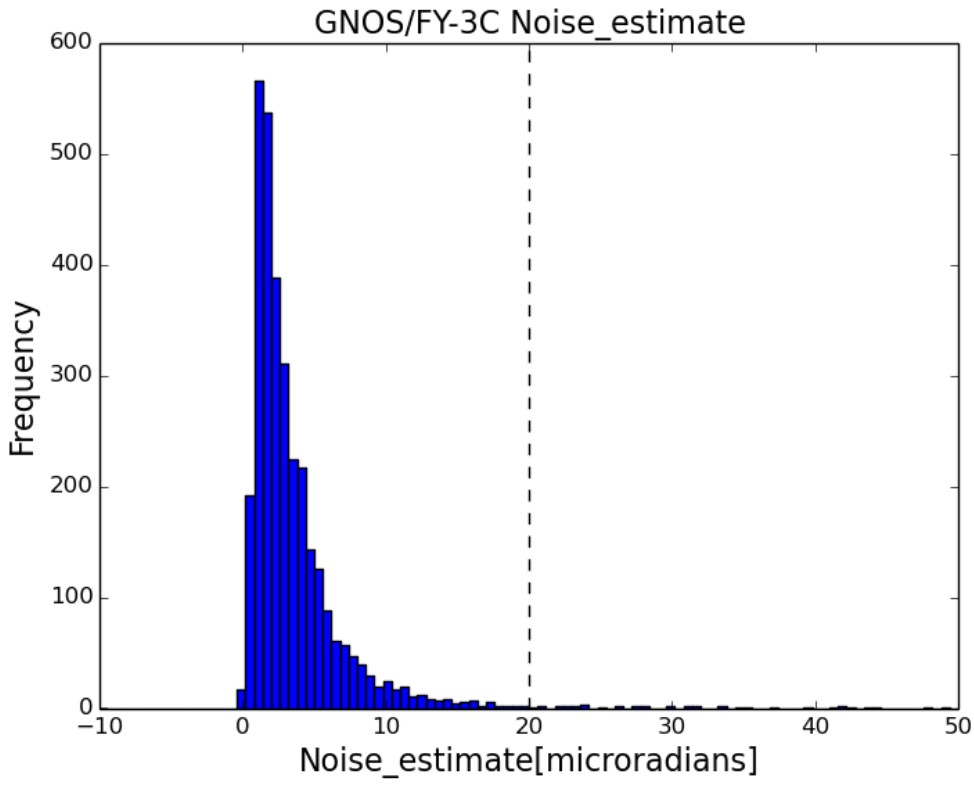

2    Figure 10. The histogram of the *noise_estimate* parameter using seven days of data

3    from 16th Feb. to 22nd Feb 2017

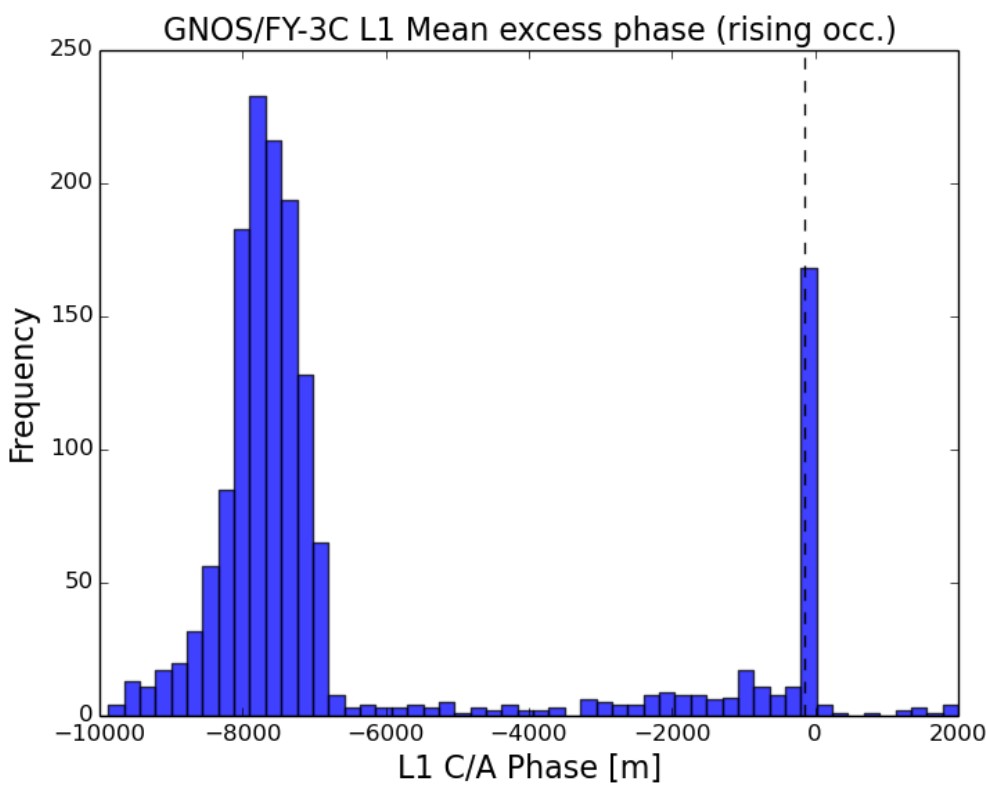

3  Figure 11. The histograms of L1 mean excess phase for the rising occultation at the
4        height of $60 - 80$ km SLTA using seven days of data from 16th Feb. to 22nd
5                                    Feb.2017.

7                                      .

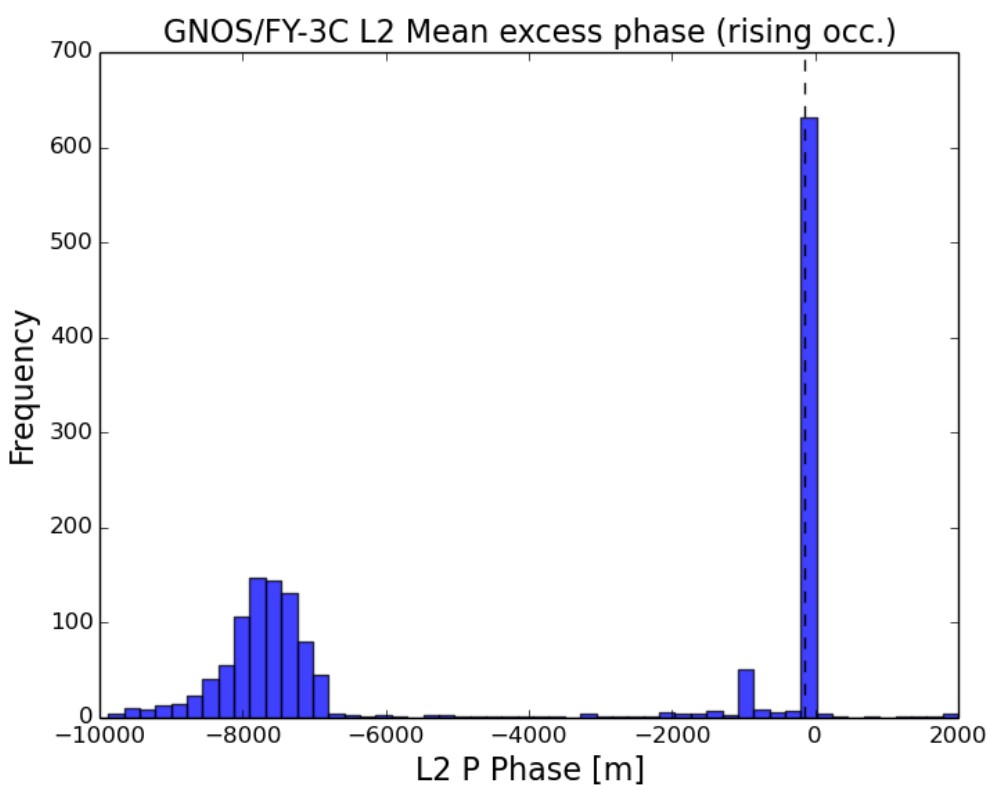

Figure 12. The histograms of L2 mean excess phase for the rising occultation at the height of 60 – 80 km SLTA using seven days of data from 16th Feb. to 22nd Feb.2017.

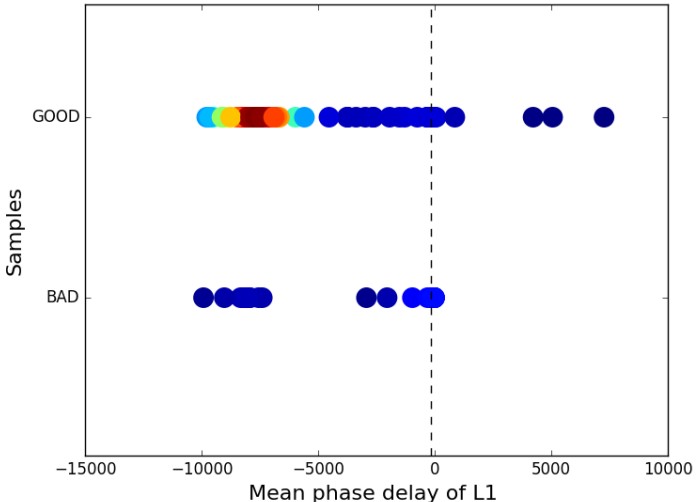

Figure 13. The L1 mean phase delay (meters) versus the good and bad samples. Different colour represents different overlap density, the dark blue is the lowest and the dark red is the highest, the colours between them show gradually higher density.

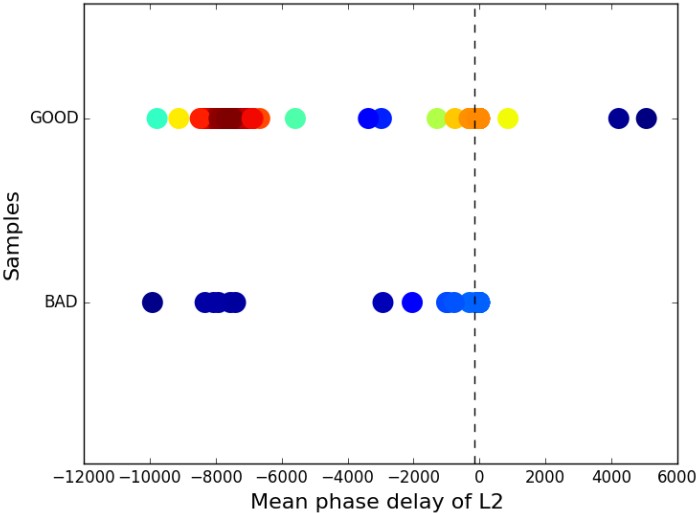

Figure 14. The L2 mean phase delay (meters) versus the good and bad samples. Different colour represents different overlap density, the dark blue is the lowest and the dark red is the highest, the colours between them show gradually higher density.

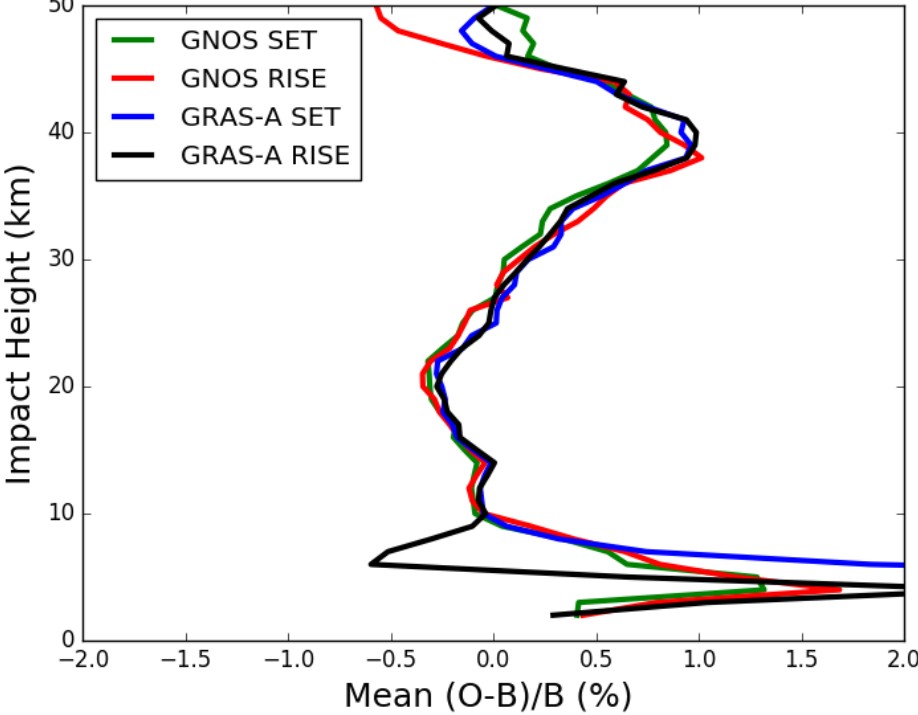

Figure 15. Global bending angle departure results, as a function of impact height, for the mean bias. The green, red, blue and black lines are representative of setting

occultation for GNOS, rising occultation for GNOS, setting occultation for GRAS and rising occultation for GRAS.

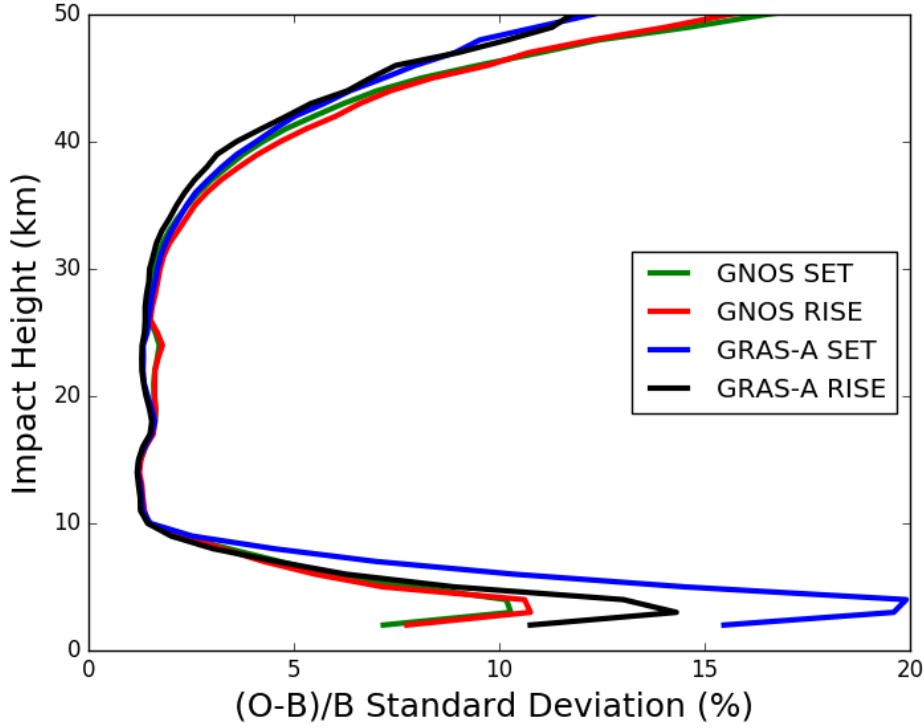

Figure 16. Global bending angle departure results, as a function of impact height, for the standard deviation. The green, red, blue and black lines are representative of setting occultation for GNOS, rising occultation for GNOS, setting occultation for GRAS and rising occultation for GRAS.