# Peer review of "Processing and quality control of FY-3C/GNOS data"

_Atmospheric Measurement Techniques, 2018_

## Referee Comment (RC1) · Anonymous Referee #2 · 26 Nov 2018

This paper is acceptable for publication in AMT with minor revisions. It describes the early data from the Chinese radio occultation sensor GNOS and shows how early problems with the L2 signal degradation were identified and largely resolved through new quality control (QC) procedures. The QC bending angle data compare favorably with GRAS data. It is encouraging to see the progress being made with the Chinese radio occultation program.

Detailed editorial comments follow. I do not need to see the revised manuscript unless significant changes are made that require further review.

1. Abstract lines 25-26. Rewrite as "GNOS bending angles and short-range ECMWF forecast bending angles....." 2. Page 2 line 2-radio occultation not capitalized 3. Page 3 line 11-suggest deleting "kinds of" 4. Page 3 line 23-poor data were filtered out...

[Figure]

(data plural) 5. Page 4 line 5-suggest deleting "in this work" 6. Page 4 line 7- are now assimilated. . .. 7. Page 4 line 24-affected not effected 8. Page 5 line 1-suggest deleting "the" before complicated Line 6-. . .not as good as that of L1 Line 7-Fig. 2 is difficult to read. Labels are too small and bars are too large, making figure out of proportion. Also, the abscissa and ordinate need labels. Line 10-can you provide references to say what is reasonable and what is not? Figs. 3 and 4-labels and legends need to be made larger 9. Page 6 line 7-Is Culverwell and Healy really unpublished if it is a ROM SAF report? I suggest deleting (unpublished). 10. Figures 5 and 6-I know it is arbitrary, but it might be better to put the "before" panel on the left and the "after"panel on the right side of the figures. Also the labels and legends should be larger. 11. Page 6 line 13: no period after Zeng 12. Page 7 line 14-reword to "L2 bending angles are very different from the L1 bending angles before correction." Line 16-reword to "..both the L1 and LC bending angles." 13. The labels in Figs. 7 and 8 are a good size. Use this size in all the figures. 14. Fig. 9-labels should be larger. 15. Page 9 line 4-"noise_estimate" is sort of a 'clunky' name. Can you come up with a symbol or shorter name? Also, the left side of (4.1) is not quite the same as the name in the text. Line 26-additional rather than extra 16. Page 10 lines 22-27-this discussion is confusing. It sounds like the % of "bad" profiles increases from 9.7% to 11.1% after QC. Please clarify and rewrite this paragraph. 17. Page 12 line 9-when the L2 signal. . . Line 13-see comment #9 above Line 26-. . .by comparing with the background bending angles computed form the operational ECMWF forecasts. 18. Page 13 line 10-reword to "We express our appreciation to. . .."

END OF COMMENTS

---

## Referee Comment (RC2) · Anonymous Referee #1 · 7 Dec 2018

**Review of paper "Processing and quality control with FY3C/GNOS data used in numerical weather prediction applications" by Mi Liao, Sean Healy, Peng Zhang**

**General Remarks**

The paper presents an interesting material on the processing of the radio occultation data acquired by the GNOS instrument. However, due to the way the material is presented, the paper requires revising. When speaking about any new approaches, the author must cite the old ones and explain, why they fail. Currently, this is not done, and, therefore, the reference of the previous work is insufficient.

**Specific Comments**

Page 2 (lines 24-30), page 3 (lines 1–2): *As with the pre-existing GPS-RO sounders…, the raw observations from GNOS consist of phase and signal to noise ratio (SNR) measurements. In addition, auxiliary information provided by the International GNSS Service (IGS), such as the GPS precise orbits, clock files, Earth orientation parameters, and the coordinates and measurements of the 1 ground stations, are also needed.*
What about the navigation bits? Does Beidou have navigation bits, similar to GPS/GLONASS ones? If so, are they also provided for the precise demodulation?

Page 3 (line 26): *if they exceed the three sigma from a statistical point of view.*
… if they exceed 3 times the standard deviation. How is the standard deviation defined?

Page 4 (lines 4–6): *Therefore in this work we developed and tested a new L2 bending angle extrapolation method for GNOS data, and implemented it in ROPP.*
Once speaking about a "new" method of the L2 extrapolation, one must cite the papers describing the "old" extrapolation technique.
1. M. E. Gorbunov, K. B. Lauritsen, A. Rhodin, M. Tomassini, and L. Kornblueh, Analysis of the CHAMP Experimental Data on Radio-Occultation Sounding of the Earth's Atmosphere, Izvestiya, Atmospheric and Oceanic Physics, 2005, V. 41, No. 6, 2005, p. 726–740.
2. M. E. Gorbunov, K. B. Lauritsen, A. Rhodin, M. Tomassini, L. Kornblueh, Radio holographic filtering, error estimation, and quality control of radio occultation data, Journal of Geophysical Research, 2006, V. 111, No. D10, D10105, doi: 10.1029/2005JD006427.

There may also be some other publications on this topic. These papers cited, the differences between the old and new approaches must be discussed. Is it the "old" extrapolation method that the authors call the "ROPP extrapolation"? Or does ROPP use a different method? What is the reason of the failure of the old extrapolation technique for the GNOS data?

Page 8 (line 15–21): *Gorbunov [2002] proposed a QC procedure in terms of the analysis of the amplitude of the RO data transformed by the Canonical Transform (CT) or the Full Spectrum Inversion (FSI) method, which is useful to catch the corrupted data because of phase lock loop failures. Beyerle et al. [2004] also suggested a QC approach to reject the RO observations ruined by ionospheric disturbances according to a parameter R defined by the phase delay of L1 and L2 signal.*
There are some more papers on QC. See the above references. See also the following papers:
3. Zou, X. & Zeng, Z. (2006), 'A quality control procedure for GPS radio occultation data', *J. Geophys. Res.* **111**, D02112.
4. Liu, H.; Kuo, Y.-H.; Sokolovskiy, S.; Zou, X.; Zeng, Z.; Hsiao, L.-F. & Ruston, B. C. (2018), 'A quality control procedure based on bending angle measurement uncertainty for

radio occultation data assimilation in the tropical lower troposphere', *J. Atmos. Oceanic Technol.* **35**(10), 2117--2131.

The paper by Zou and Zeng is in the reference list, but is not discussed nor referenced in the text. Please provide a comparative analysis of the old and new QC methods with the explanation of why the old QC methods are not sufficient for your data analysis. In particular, will the "badness score" introduced by Gorbunov et al. and successfully applied for CHAMP, COSMIC, METOP and other observations, be also useful for the FY3C/GNOS data analysis? If not, why?

Page 9 (lines 3–7): *The physical meaning of noise_estimate is easy to understand.*
What is easy to understand is the fact that $\Delta\alpha$ is restricted to be close enough to its estimate obtained from a simple ionospheric model. Nevertheless, it is a good idea for the authors to explicitly mention this rather than appeal that something is "easy to understand". Still, some questions remain. Does $n$ in formula (4.1) stay for refractivity of number of data? Number of data is definitely missing somewhere, because the sum in this formula needs to be normalized by the number of data. If $n$ is refractivity, at what height is it taken? Provide explanations or definition regarding $n$.

Page 11 (lines 21–22): *The GRAS standard deviations are worse in the troposphere might due to sampling; essentially GRAS is able to measure more difficult cases.*
This statement needs more explanation. What are "more difficult cases"? Do they mostly occur in tropics? Can the authors provide any examples? Is it possible to evaluate a regionalized statistics (tropics, mid-, and polar latitudes)?

---

## Referee Comment (RC3) · Anonymous Referee #3 · 12 Dec 2018

**Date: December 11, 2018**

Manuscript #: amt-2018-271
Manuscript title: ***Processing and quality control of FY-3C/GNOS data used in numerical weather prediction applications***

**Brief Summary of the Manuscript**
This manuscript presents an approach to correct for L2 signal degradation of the GNOS receiver, as well as it introduces quality control (QC) checks to evaluate this new approach in bending angle profiles. Finally, the manuscript presents comparison statistics between the GNOS L2-corrected profiles with ECMWF. Despite the authors' efforts, there are serious concerns regarding the physical-mathematical interpretation of the new approach and in extent of the results presented in this manuscript. Based on the comments below, I recommend rejection of this manuscript.

**Major Comments:**
1) ***Introduction.*** The manuscript lacks motivation. Since the authors present a new methodology to correct the L2 signal bending, the "old" ROPP L2 signal correction approach should be described. Additionally, the differences between the "old" and the "new" approaches should be emphasized and discussed in detail. Currently, the reader cannot understand why the current ROPP approach does not work for the GNOS retrievals and all relevant references are missing.

2) ***Introduction: P. 3; Line 30.*** "These biases are not seen with other RO missions." Yes, the L2 signal is weaker than the L1 signal. However, other RO missions do not lose L2 signal tracking that much high up in the neutral atmosphere. The authors should explain why GNOS loses L2 signal tracking in the stratosphere at ~ 20 km, unlike all other RO missions. The authors state that the most prominent quality issue was the large departures biases, in the vertical range of 5 – 30 km. This altitude covers the middle troposphere up to the middle stratosphere. Then, within this context, if GNOS loses 30% of the profiles below 20 km (see P. 5; Line 11), then the authors should explain how does GNOS contribute to Numerical Weather Prediction (NWP) and specify the most effective altitude range of the GNOS RO profiles.

3) **New L2 extrapolation**: Equation (3.4) states that the bending angle in L2 frequency equals the bending angle in L1 frequency plus a correction factor, which is proportional to the ionospheric TEC. The problem in Equation (3.3) is that it is derived using Equation (3.2), which is valid only for ionospheric bending and not for neutral atmosphere bending, as specifically mentioned in *Culverwell and Healy* (2015). Within the neutral atmosphere the ionospheric bending becomes negligible and the signal bending at tropospheric and stratospheric altitudes has an exponential dependency on the impact parameter – different than Equation (3.2). Therefore, how could the authors apply Equation (3.2) to correct for the L2 bending angle within the neutral atmosphere using bending angle approximations derived for ionospheric bending only – particularly when applying this method from the lowest altitude the L2 signal is lost and 20 km up with a maximum upper limit of 70 km that is around the bottomside of the ionospheric D layer?

4) **Equation (4.1):** The $X_{so}$ is estimated from the least squares fit between the observed L1 and L2 bending angles. Then again, the new *noise_estimate* the authors introduce defines a new statistical metric based on how close the $X_{so}$ is to the observed L1 and L2 bending angle difference. But, the $X_{so}$ was estimated in Equation (3.4) to fit the minimum bending angle difference in L1 and L2. This *noise_estimate* appears to be misleading, without physical underpinning and with an over-fitting nature that beats down the scatter. Additionally, P. 9; Line 8: "The physical meaning of *noise_estimate* is easy to understand." Is not easy to understand and the authors should explain the rationale of defining it, because the $X_{so}$ has already been estimated well via Equation (3.4). Also, how do the authors decide on the 20 microradiances as the threshold value?

5) **Section 4.2:** The authours do not explain why is it necessary to monitor the performance of GNOS mean L1 and L2 phase delays in the height interval of 60 to 80 km. Also, why the mean phase and not the phase variation with altitude within this height range? What GNOS product is assimilated in NWP models and how does monitoring the 60-80 km phase delays help us to QC the profile below?

6) **P. 10; Line 21:** "*...these have been tested with one day of data...*" The statistical sampling used in the determination of the statistical performance of the QC methods is low and does not represent the statistical performance of the GNOS profiles around the globe and under different seasons.

7) **Section 5:** The authours explanation of the 15% disagreement between the GNOS and GRAS profiles below 10 km is inadequate. Ideally, collocated profiles between GNOS and GRAS should be used to quantify the degree of agreement or disagreement. However, if there are not enough collocated profiles between July 6 and August 2, 2018, perhaps the authours could use the entire time period GNOS provides RO profiles and if there are still not enough collocated profiles the authours could bin their profiles either into latitude sectors or seasons and then compare with GRAS to create an ensemble study to greatly increase the statistical sampling. ***The results represent a limited statistical sampling to support the authours' claims.***

**Minor Comments:**
 a) **P. 2; Line 16:** "...velocity and anti-velocity antennas..." Do you mean fore and aft antennas?
 b) **P. 2; Line 19:** What is the GNOS inclination in Table 1?
 c) **P. 2; Line 17:** Is BDS global or region constellation. Mention geographic restrictions of RO.
 d) **P. 3; Line 22:** "...departure statistics..." From what?
 e) **P. 3; Line 25:** Why more than 20% levels of the profile? How was this threshold selected? Explain.
 f) **P. 4; Line 10:** What is the most effective altitude range that GNOS provides the best RO profiles and explain how this information is used in NWP and how does it improve NWP. Include references to support claims.
 g) **P. 4; Line 14:** "...may..." replace with "...could be..."
 h) **P. 5; Line 11:** Is this L2 signal loss at 20 km normal? Usually L2 signal is lost in the middle troposphere which is about 5 km. Explain.
 i) **P. 5; Line 27:** "...consistency..." replace with "...agreement..."
 j) **P. 6; Line 5:** Define "obvious errors".
 k) **P. 6; Line 9:** Define "other profiles".
 l) **P. 6; Line 11:** This definition of the ionosphere is crude, general, and unrealistic. Usually, the ionosphere is represented with multiple Chapman profiles with different scale heights. Mathematically, the Dirac function obtains a value of 0 at altitudes outside a very small neighbourhood of the peak height.
 m) **P. 6; Line 27:** Why the peak height is 300 km? What led to this selection? The rule of thumbs says that per 100 km different in ionospheric shell height leads to 1 TECU error in the ionospheric total electron content. How sensitive is the estimation of $X_{so}$ to the ionospheric TEC?
 n) **P. 7; Equation (3.4):** This equation describes the ionospheric bending angle and not the neutral atmosphere. How can the authors apply this equation to correct for the L2 bending in the neutral atmosphere?

---

## Short Comment (SC1) · 14 Dec 2018

**Response to Major Comment 3) by Reviewer 3.**

**3) New L2 extrapolation**

The reviewer makes a number of important points, which we will respond to more fully at a later date. However, we believe that the main comments, regarding the questionable physical-mathematical basis of the new L2 extrapolation method, are incorrect. They are probably a result of a lack of clarity in the submitted manuscript, regarding the key assumptions.

In the neutral atmosphere, the contribution of the ionospheric bending to the total bending falls in a fractional sense, because the neutral bending increases exponentially towards the surface. However, the ionospheric bending is not "negligible", and it cannot be ignored in NWP applications.

An assumption made in most GPS-RO ionospheric correction bending (e.g., Vorob'ev and Krasil'nikova, 1994) and extrapolation routines (e.g., Zeng et al, 2016) is that the total bending angle for frequency $k$, $\alpha_k$, can be written as the sum of the neutral bending, $\alpha_n$, plus a frequency dependent ionospheric term, $\alpha_{i,k}$, which scales with frequency as, $1/f^2$ . Hence,

$$\alpha_k = \alpha_n + \alpha_{i,k}$$

where $k = (1,2)$ for the $f_1$ and $f_2$ frequency, respectively. This decomposition is the basis of the standard linear ionospheric correction (for a common impact parameter),

$$\alpha_n = \alpha_1 + \frac{f_2^2}{f_1^2 - f_2^2}(\alpha_1 - \alpha_2)$$

used to estimate the neutral bending (e.g., Vorob'ev and Krasil'nikova, 1994). Note that the difference $\alpha_1 - \alpha_2$ should be independent of the neutral atmosphere, $\alpha_{i,1} - \alpha_{i,2}$, and it will only vary slowly with height (e.g., see Figures 2 and 3, Zeng et al, 2016). "Residual" ionospheric errors caused by non-linear terms can also be accounted for (e.g., Healy and Culverwell, 2015), but they are typically a few tenths of a microradian, and they are not of importance here.

In this paper, we also assume that the total bending can be approximated by the sum of the neutral and ionospheric terms, and then we use a simple ionospheric model to fit the *neutral free* bending angle differences, $\alpha_1 - \alpha_2$, over a 20 km vertical interval. These fitting parameters are then used to extrapolate the $\alpha_1 - \alpha_2$ values where the L2 signals are lost or significantly degraded.

Therefore, assumptions made are entirely consistent with the standard ionospheric correction, but we accept that this could be made clearer in the original manuscript.

References

Healy, S. B. and Culverwell, I. D.: A modification to the standard ionospheric correction method used in GPS radio occultation, Atmos. Meas. Tech., 8, 3385-3393, https://doi.org/10.5194/amt-8-3385-2015, 2015.

Vorob'ev, V. V. and Krasil'nikova, G. K.: Estimation of the accuracy of the atmospheric refractive index recovery from dopplershift measurements at frequencies used in the NAVSTAR system,USSR Phys. Atmos. Ocean, Engl. Transl., 29, 602–609, 1994.

Zeng, Z., Sokolovskiy, S., Schreiner, W., Hunt, D., Lin, J., and Kuo, Y.-H.: Ionospheric correction of GPS radio occultation data in the troposphere, Atmos. Meas. Tech., 9, 335-346, https://doi.org/10.5194/amt-9-335-2016, 2016.

---

## Short Comment (SC2) · 18 Dec 2018

Statistics for matched occultations are routinely available from the ROM SAF web pages.

See,

http://www.romsaf.org/monitoring/matched.php

An example for GNOS versus Metop-A GRAS is attached.

The GNOS data presented on these pages is processed with the method outlined in the paper. However, we do not believe that the matched occultation statistics provide any additional information, relative to the bending angle departure statistics computed

with an accurate short-range forecast.

[Figure]

BA Stats of co-located BA (3.0h, 300.0km):
Metop-A processed by DMI vs
FY-3C processed by CMA

QC applied and outliers removed
1210 matches after QC.
Data from 19/11/18 to 18/12/18

**Fig. 1.** GNOS vs GRAS matched statistics from the ROM SAF web pages.

---

## Short Comment (SC3) · 20 Dec 2018

We appreciate that the reviewer provides such valuable comments.

**Responses to the specific comments**

*1. Page 2 (lines 24-30), page 3 (lines 1–2): As with the pre-existing GPS-RO sounders…, the raw observations from GNOS consist of phase and signal to noise ratio (SNR) measurements. In addition, auxiliary information provided by the International GNSS Service (IGS), such as the GPS precise orbits, clock files, Earth orientation parameters, and the coordinates and measurements of the 1 ground stations, are also needed.*

*What about the navigation bits? Does Beidou have navigation bits, similar to GPS/GLONASS ones? If so, are they also provided for the precise demodulation?*

A: The navigation bits contain information concerning the satellite clock, the satellite orbit, the satellite health status, and various other data. Beidou have navigation bits too. IGS provides the Beidou navigation bits, but not in near real time. The timeliness can be about 7 days.

*2. Page 3 (line 26): if they exceed the three sigma from a statistical point of view.*

*… if they exceed 3 times the standard deviation. How is the standard deviation defined?*

A: GNOS data is compared to background data, e.g. ECMWF reanalysis. The standard deviation is defined as $std = \frac{\sqrt{\sum(x_i - \bar{x})^2}}{n}$, $n$ is number, $x_i = \left(\frac{O-B}{B}\right) * 100\%$, $\bar{x}$ is the average of $x_i$.

*3. Page 4 (lines 4–6): Therefore in this work we developed and tested a new L2 bending angle extrapolation method for GNOS data, and implemented it in ROPP.*

*Once speaking about a "new" method of the L2 extrapolation, one must cite the papers describing the "old" extrapolation technique.*

*There may also be some other publications on this topic. These papers cited, the differences between the old and new approaches must be discussed. Is it the "old" extrapolation method that the authors call the "ROPP extrapolation"? Or does ROPP use a different method? What is the reason of the failure of the old extrapolation technique for the GNOS data?*

A: We agree that the "standard" ROPP should be described in more detail in the revised manuscript.

In the context of the difficulties processing GNOS data, ROPP includes a pre-processing step in order to correct degraded L2 data. The approach is based on Gorbunov et al (2005,2006), and it is used routinely for other GPS-RO missions. Briefly, smoothed L1 and L2 bending angle and impact parameters are computed. An impact height, PC, above which the L2 data is considered reliable, is estimated using an empirical "badness score". The empirical badness score at time $t$, is defined as,

$$Q(t) = \left(\frac{abs(\overline{p_1(t)} - \overline{p_2(t)})}{\Delta p_a} + \frac{\delta p_2(t)}{\Delta p_b}\right)^2$$

where $\delta p_2$ is a measure of the width of the L2 spectrum, $\overline{p_1(t)}$ and $\overline{p_2(t)}$ are the L1 and L2 impact parameters, respectively, computed from smoothed timeseries, $\Delta p_a$ =200 m and $\Delta p_b$ =150 m (See also, Eq. 11 Gorbunov et al, 2006 for a slightly modified form). The largest $Q(t)$

value in the impact height interval between 15 km to 50 km is stored as the badness score for the occultation, potentially for quality control purposes.

The mean L1 and L2 bending angle and impact parameters are then computed in a 2 km impact parameter interval directly above PC. Simulated L2 bending angles and impact parameters are computed by adding the mean (L2-L1) differences to both the L1 bending angle and impact parameter values, using the data in the 2 km interval. Simulated L2 and L1 phase values are then computed from these bending angles. Corrected L2 excess phase values are computed by merging the observed L2 phase above PC, with the simulated values below PC, using a smooth transition over 2 km, centred on PC. The corrected L2 phase values are subsequently used in the wave optics processing of the L2 signals.

A difficulty with the GNOS processing is related to determining the impact height PC, used for both the computation of the mean L1 and L2 differences, and defining the transition between observed and modelled L2 phase values. Although the "badness score" is used to determine PC, PC also has a maximum value (20 km). This is defined as the wave optics processing height (25 km) minus a 5 km "safety border". Therefore, the mean bending angles and impact parameters used in the L2-L1 correction can only be computed in a 2 km interval up to a maximum impact height of 22 km. Unfortunately, this is not high enough for GNOS L2 signals, with the result that the mean L2-L1 bending angle and impact parameters computed in the 2 km interval above PC are corrupted.

M. E. Gorbunov, K. B. Lauritsen, A. Rodin, M. Tomassini, and L. Kornblueh (2005), Analysis of the CHAMP Experimental Data on Radio-Occultation Sounding of the Earth's Atmosphere, Izvestiya, Atmospheric and Oceanic Physics, 41, No. 6, 726–740.

Gorbunov, M. E., K. B. Lauritsen, A. Rhodin, M. Tomassini, and L. Kornblueh (2006), Radio holographic filtering, error estimation, and quality control of radio occultation data, J. Geophys. Res., 111, D10105, doi:10.1029/2005JD006427.

4. *The paper by Zou and Zeng is in the reference list, but is not discussed nor referenced in the text. Please provide a comparative analysis of the old and new QC methods with the explanation of why the old QC methods are not sufficient for your data analysis. In particular, will the "badness score" introduced by Gorbunov et al. and successfully applied for CHAMP, COSMIC, METOP and other observations, be also useful for the FY3C/GNOS data analysis? If not, why?*

A: Thanks for the comments. More references will be cited and discussed in the revised manuscript. Originally, we'd like to find out a method to identify the quality of GNOS profiles based on physical meaning and without using background data, just as the "badness score". When we look at the performance of "badness score", it is not suitable for GNOS (see fig1). The values of L2 badness score range from 15 to 1000 plus. The reason might be related to some empirical parameters. Other missions work well using "badness score" since the lowest SLTA of L2 is low enough. When discussed with scientists from EUMETSAT, GRAS can get down to 15km for more than 90%. But it is not the case for GNOS. Only 70% of L2 can be reached below 20km. So the noise_estimate parameter is used as a quality indicator, which could

show the performance of L2 extrapolation.

[Figure]

Fig 1. The cases of L2 badness score fail our QC and pass our QC

5. *Page 9 (lines 3–7): The physical meaning of noise_estimate is easy to understand.*
*What is easy to understand is the fact that $\Delta\alpha$ is restricted to be close enough to its estimate obtained from a simple ionospheric model. Nevertheless, it is a good idea for the authors to explicitly mention this rather than appeal that something is "easy to understand". Still, some questions remain. Does n in formula (4.1) stay for refractivity of number of data? Number of data is definitely missing somewhere, because the sum in this formula needs to be normalized by the number of data. If n is refractivity, at what height is it taken? Provide explanations or definition regarding n.*
A: Thanks for the suggestion. n in formula 4.1 is the number of data. This will be fixed in the revised manuscript.

6. *Page 11 (lines 21–22): The GRAS standard deviations are worse in the troposphere might due to sampling; essentially GRAS is able to measure more difficult cases.*
*This statement needs more explanation. What are "more difficult cases"? Do they mostly occur in tropics? Can the authors provide any examples? Is it possible to evaluate a regionalized statistics (tropics, mid-, and polar latitudes)?*
A: The comparison between GRAS and GNOS is not the most important part of the manuscript, thus a general remark is made. However, the work is worthy to be done. We'll see if it is possible to add statistics related to the two data.

---

## Short Comment (SC4) · 20 Dec 2018

We appreciate the comments from the anonymous referee. Those comments will be accepted item by item in the revised manuscript.
* * *

---

## Short Comment (SC5) · 20 Dec 2018

We have attached departure statistics plots for the tropics and south polar region (lat < -65). Fig 1 and 2 are for tropical bias and standard deviation, respectively. Figure 3 and 4 are the bias and standard deviation for the south pole. The plots are for the period July 6- Aug 2, 2018, in order to be consistent with Figures 13-14. It is worth noting that these statistics have been derived from the operational ECMWF processing, and both the GRAS and GNOS have been subjected to the same quality control criteria. Essentially, departures greater than $\sim$10 times the assumed observation error are removed, and the same error model is used for GNOS and GRAS.

The purpose of these plots is to show that the departure statistics for GNOS and Metop-

[Figure]

A GRAS are similar. In general, the GRAS standard deviations are larger than GNOS in the troposphere. Further, it is known from operational monitoring that departure statistics for GRAS setting are larger than GRAS rising. The hypothesis for this GRAS result is that setting occultations penetrate more deeply in moist atmospheres. The results in south polar region shown here suggest that it may not be the full story, and this warrants further investigations. However, the primary focus of this paper is the quality of GNOS data with the new processing, and we believe we have shown that it is comparable to GRAS.

———————————————————

[Figure]

Fig. 1. Bias in tropics

Fig. 2. Standard deviation in tropics

[Figure]

**Fig. 3.** Bias S.Pole

[Figure]

**Fig. 4.** Standard deviation S. Pole

---

## Short Comment (SC6) · 7 Jan 2019

We thank reviewer for the comments.

**Responses to the specific comments**

*1) Introduction. The manuscript lacks motivation. Since the authors present a new methodology to correct the L2 signal bending, the "old" ROPP L2 signal correction approach should be described. Additionally, the differences between the "old" and the "new" approaches should be emphasized and discussed in detail. Currently, the reader cannot understand why the current ROPP approach does not work for the GNOS retrievals and all relevant references are missing.*

A: Thank you for pointing out the problem. We will add the relevant references about the old and approaches to clarify the GNOS retrievals in the revised manuscript.

*2) Introduction: P. 3; Line 30. "These biases are not seen with other RO missions." Yes, the L2 signal is weaker than the L1 signal. However, other RO missions do not lose L2 signal tracking that much high up in the neutral atmosphere. The authors should explain why GNOS loses L2 signal tracking in the stratosphere at ~ 20 km, unlike all other RO missions. The authors state that the most prominent quality issue was the large departures biases, in the vertical range of 5 – 30 km. This altitude covers the middle troposphere up to the middle stratosphere. Then, within this context, if GNOS loses 30% of the profiles below 20 km (see P. 5; Line 11), then the authors should explain how does GNOS contribute to Numerical Weather Prediction (NWP) and specify the most effective altitude range of the GNOS RO profiles.*

A: The reason for GNOS losing L2 signal tracking is that GNOS has a lower SNR compared to other missions. Additionally, the GNOS antenna is smaller and not well located on the satellite. Consequently, we have to use additional cables, which results in a larger decrease of SNR than expected. Scientists from EUMETSAT confirmed that GRAS can get down to 15 km for more than 90% of the cases, but it is not the case for GNOS. Only 70% of L2 can reach below 20km. However, note that GNOS on FY3C is just the first Chinese GPS-RO mission. For the second satellite, FY3D, GNOS has more antenna units and in turn, has higher SNR than FY3C. Thus, the L2 signal tracking gets better. The proportion of the large departures biases in FY3D is smaller than in FY3C as well.

[Figure]

[Figure]

[Figure]

[Figure]

Figure 1

It is true that GNOS initially lost 30% of the profiles below 20 km, but that was before applying the new L2 extrapolation method outlined in the paper. After adopting the new method, we can process more GNOS profiles successfully. .

Regarding the impact on numerical weather prediction, GNOS was tested in the ECMWF assimilation system for the period November 23, 2017 to March 5,2018, prior to operational assimilation in the ECMWF system in March 2018. GNOS is assimilated operationally in the impact height interval from 8 km to 50 km in the extra-tropics, and from 10 km to 50 km in the

tropics. Although the medium-range forecast scores were generally neutral, in the short-range, the assimilation of GNOS data clearly improved the fit to other GPS-RO data, such as Metop GRAS A,B GRAS, COSMIC-6 etc. Figure 2 shows the improvement in the GPS-RO departure statistics for short-range forecasts when GNOS data is assimilated. This Figure could be added the final manuscript, but the main focus of the paper is how the current operational FY3C GNOS data is processed, rather than the impact in NWP systems.

[Figure]

Figure 2: The percentage change in the GPS-RO departure statistics as a result of assimilating the GNOS measurements. The change in the standard deviation of the background (o-b) departures are on the right, and the analysis (o-a) departures are on the left. The statistics are globally averaged, and the dotted lines indicated 95 % statistical significance. Values less than 100 % on the left hand side indicate that the short-range forecasts fit the other GPS-RO data more closely as a result of assimilating GNOS.

*5) Section 4.2: The authors do not explain why is it necessary to monitor the performance of GNOS mean L1 and L2 phase delays in the height interval of 60 to 80 km. Also, why the mean phase and not the phase variation with altitude within this height range? What GNOS product is assimilated in NWP models and how does monitoring the 60-80 km phase delays help us to QC the profile below?*

A: We take these phase delays as one of QC factors because empirically it was found to determine the performance of GNOS when compared with reanalysis data. When encountering the bad profiles, the rising L1 and L2 mean phase delays have small values. The result is only based on FY3C. Subsequently, when we look at FY3D, this phenomenon disappears. Thus this factor is not a general one. We are considering cutting this part of from the manuscript.

6) P. 10; Line 21: "...these have been tested with one day of data..." The statistical sampling used in the determination of the statistical performance of the QC methods is low and does not represent the statistical performance of the GNOS profiles around the globe and under different seasons.

A: One day of data was used to initially estimate the various QC parameters and then these were tested over longer periods. Clearly, the new L2 extrapolation method is rather effective at eliminating the large errors for the longer period, globally (See Figure 13,14)The plot shown here is just an example.

Minor comments:
a) P. 2; Line 16: "...velocity and anti-velocity antennas..." Do you mean fore and aft antennas?
A: Yes
b) P. 2; Line 19: What is the GNOS inclination in Table 1?
A: The inclination of FY3C/GNOS is 98.75 °
c) P. 2; Line 17: Is BDS global or region constellation. Mention geographic restrictions of RO.
A: BDS both has global and region constellation. The distribution of BDS RO can be shown as follows, also it can be refered to Mi Liao et al.,2016

[Figure]

Figure 3. Map of the GNOS BDS occultation coverage from
1 November to 31 December 2013, with a total of 4648 samples.
Different colours indicate different penetration depths.

[Figure]

Figure 4. Map of the GNOS BDS occultation coverage

Different colours indicate different constellations. MEO have the same altitude as GPS.

d) P. 3; Line 22: "...departure statistics..." From what?

A: From background data, such as forecast data.

e) P. 3; Line 25: Why more than 20% levels of the profile? How was this threshold selected? Explain.

A: Compared with background data, the bad profiles are defined as the mean biases greater than 10% (100*(O-B)/B) from 5km to 30 km. As we know that the bias of RO at that height is about 1% in normal case. If the threshold is set as 10%, the large departure profiles can be identified.

f) P. 4; Line 10: What is the most effective altitude range that GNOS provides the best RO profiles and explain how this information is used in NWP and how does it improve NWP. Include references to support claims.

A: Currently, there are no published papers talking about the GNOS in NWP. Only some technical reports from personal communications. However, see Figure 2 above.

g) P. 4; Line 14: "...may..." replace with "...could be..."

A: Fine.

h) P. 5; Line 11: Is this L2 signal loss at 20 km normal? Usually L2 signal is lost in the middle troposphere which is about 5 km. Explain.

A: This can be seen from my reply to your second major comment.

i) P. 5; Line 27: "...consistency..." replace with "...agreement..."

A: Fine.

j) P. 6; Line 5: Define "obvious errors".

A: Fine.

k) P. 6; Line 9: Define "other profiles".

A: Fine.

---

## Author Comment (AC1) · 3 Feb 2019

**Responses to the specific comments**

1. Abstract lines 25-26. Rewrite as "GNOS bending angles and short-range ECMWF forecast bending angles: : :.."

A: Done

2. Page 2 line 2-radio occultation not capitalized.

A: Done

3. Page 3 line 11-suggest deleting "kinds of".

A: Done

4. Page 3 line 23-poor data were filtered out…(data plural)

A: Done

5. Page 4 line 5-suggest deleting "in this work"

A: Done

6. Page 4 line 7- are now assimilated...

A: Done

7. Page 4 line 24-affected not effected.

A: Done

8. Page 5 line 1-suggest deleting "the" before complicated Line 6-…not as good as that of L1 Line 7-Fig. 2 is difficult to read. Labels are too small and bars are too large, making figure out of proportion. Also, the abscissa and ordinate need labels. Line 10-can you provide references to say what is reasonable and what is not? Figs. 3 and 4-labels and legends need to be made larger.

A: Done. The figures have been replotted.

[Figure]

Figure 2. Ratio of different SLTA of the L1 C/A and L2 P for the rising and setting occultations, statistics result is from 28th Jan to 2nd Feb. 2017.

[Figure]

Figure 3.

[Figure]

Figure 4.

9. Page 6 line 7-Is Culverwell and Healy really unpublished if it is a ROM SAF report? I suggest deleting (unpublished).

A: This work is based on the work of Culverwell and Healy, which is an important source. Although it hasn't been published, we'd like to refer it as an informal report.

10. Figures 5 and 6-I know it is arbitrary, but it might be better to put the "before" panel on the left and the "after"panel on the right side of the figures. Also the labels and legends should be larger.

A: Done.

[Figure]

Figure 5. Case1: the bending angle of L2 (red), L1 (green) and LC (yellow) before (right) and after (left) correction.

[Figure]

Figure 6. The same as Figure 5 but for Case 2.

11. Page 6 line 13: no period after Zeng.
A: Done.
12. Page 7 line 14-reword to "L2 bending angles are very different from the L1 bending angles before correction." Line 16-reword to "..both the L1 and LC bending angles."
A: Done.
13. The labels in Figs. 7 and 8 are a good size. Use this size in all the figures.
A: Done.
14. Fig. 9-labels should be larger.
A: Done.
15. Page 9 line 4-"noise_estimate" is sort of a 'clunky' name. Can you come up with a symbol or shorter name? Also, the left side of (4.1) is not quite the same as the name in the text. Line 26-additional rather than extra.

A: Done. We will replace the "noise_estimate" as $\theta_\alpha$.

16. Page 10 lines 22-27-this discussion is confusing. It sounds like the %of "bad" profiles increases from 9.7% to 11.1% after QC. Please clarify and rewrite this paragraph.

A: Combining other reviewers' comments, this part is not necessary and confusing; we decide to delete this part.

17. Page 12 line 9-when the L2 signal… Line 13-see comment #9 above Line 26-…by comparing with the background bending angles computed form the operational ECMWF forecasts.

A: Done.

18. Page 13 line 10-reword to "We express our appreciation to…"

A: Done.

---

## Author Response (AR1)

**For reviewer #1**

**Changes in the manuscript:**

**Considering the specific comments, all the detailed changes as the reviewer requested have been made in the track version of the manuscript, please see the related file.**

**Responses to the specific comments**

1. Abstract lines 25-26. Rewrite as "GNOS bending angles and short-range ECMWF forecast bending angles: : :.."

**Author's response:** Done

2. Page 2 line 2-radio occultation not capitalized.

**Author's response:** Done

3. Page 3 line 11-suggest deleting "kinds of".

**Author's response:** Done

4. Page 3 line 23-poor data were filtered out…(data plural)

**Author's response:** Done

5. Page 4 line 5-suggest deleting "in this work"

**Author's response:** Done

6. Page 4 line 7- are now assimilated...

**Author's response:** Done

7. Page 4 line 24-affected not effected.

**Author's response:** Done

8. Page 5 line 1-suggest deleting "the" before complicated Line 6-…not as good as that of L1 Line 7-Fig. 2 is difficult to read. Labels are too small and bars are too large, making figure out of proportion. Also, the abscissa and ordinate need labels. Line 10-can you provide references to say what is reasonable and what is not? Figs. 3 and 4-labels and legends need to be made larger.

**Author's response:** Done. The figures have been replotted.

[Figure]

Figure 2. Ratio of different SLTA of the L1 C/A and L2 P for the rising and setting occultations, statistics result is from 28th Jan to 2nd Feb. 2017.

[Figure]

Figure 3.

[Figure]

Figure 4.

9. Page 6 line 7-Is Culverwell and Healy really unpublished if it is a ROM SAF report? I suggest deleting (unpublished).

**Author's response:** This work is based on the work of Culverwell and Healy, which is an important source. Although it hasn't been published, we'd like to refer it as an informal report.

10. Figures 5 and 6-I know it is arbitrary, but it might be better to put the "before" panel on the left and the "after"panel on the right side of the figures. Also the labels and legends should be larger.

**Author's response:** Done.

[Figure]

Figure 5. Case1: the bending angle of L2 (red), L1 (green) and LC (yellow) before (right) and after (left) correction.

[Figure]

Figure 6. The same as Figure 5 but for Case 2.

11. Page 6 line 13: no period after Zeng.
**Author's response:** Done.
12. Page 7 line 14-reword to "L2 bending angles are very different from the L1 bending angles before correction." Line 16-reword to "..both the L1 and LC bending angles."
**Author's response:** Done.
13. The labels in Figs. 7 and 8 are a good size. Use this size in all the figures.
**Author's response:** Done.
14. Fig. 9-labels should be larger.
**Author's response:** Done.
15. Page 9 line 4-"noise_estimate" is sort of a 'clunky' name. Can you come up with a symbol or shorter name? Also, the left side of (4.1) is not quite the same as the name in the text. Line 26-additional rather than extra.

**Author's response:** Done. We will replace the "noise_estimate" as $\theta_\alpha$.

16. Page 10 lines 22-27-this discussion is confusing. It sounds like the %of "bad" profiles increases from 9.7% to 11.1% after QC. Please clarify and rewrite this paragraph.

**Author's response:** Combining other reviewers' comments, this part is not necessary and confusing; we decide to delete this part.

**Changes in the manuscript:** Reconsider the content of the file, we decide to keep this part. We carried out the statistics using more samples (three months of data) and reworded this part. Please see the track version from p10 to p11.

17. Page 12 line 9-when the L2 signal… Line 13-see comment #9 above Line 26-…by comparing with the background bending angles computed form the operational ECMWF forecasts.

**Author's response:** Done.

18. Page 13 line 10-reword to "We express our appreciation to…"

**Author's response:** Done.

[revised manuscript text omitted]

**For reviewer #2**

1.

**Reviewer's comment:** Page 2 (lines 24-30), page 3 (lines 1–2): As with the pre-existing GPS-RO sounders…, the raw observations from GNOS consist of phase and signal to noise ratio (SNR) measurements. In addition, auxiliary information provided by the International GNSS Service (IGS), such as the GPS precise orbits, clock files, Earth orientation parameters, and the coordinates and measurements of the 1 ground stations, are also needed.

What about the navigation bits? Does Beidou have navigation bits, similar to GPS/GLONASS ones? If so, are they also provided for the precise demodulation?

**Author's response:** The navigation bits contain information concerning the satellite clock, the satellite orbit, the satellite health status, and various other data. Beidou have navigation bits too. IGS provides the Beidou navigation bits, but not in near real time. The timeliness can be about 7 days.

**Changes in the manuscript:** This is not shown in the revised manuscript. Because we believe it's not much related to the paper.

2.

**Reviewer's comment:** Page 3 (line 26): if they exceed the three sigma from a statistical point of view.

… if they exceed 3 times the standard deviation. How is the standard deviation defined?

**Author's response:** GNOS data is compared to background data, e.g. ECMWF reanalysis. The

standard deviation is defined as $std = \frac{\sqrt{\sum(x_i - \overline{x})^2}}{n}$, $n$ is number, $x_i = \left(\frac{O-B}{B}\right) * 100\%$, $\overline{x}$ is the

average of $x_i$.

**Changes in the manuscript:** This part is overlapped with the first part of the section 2. For better elaboration, we decide to delete this part in the revised manuscript. The correction is at p3 line 22-30 in the track changes version.

3.

**Reviewer's comment:** Page 4 (lines 4–6): Therefore in this work we developed and tested a new L2 bending angle extrapolation method for GNOS data, and implemented it in ROPP.

Once speaking about a "new" method of the L2 extrapolation, one must cite the papers describing the "old" extrapolation technique.

There may also be some other publications on this topic. These papers cited, the differences between the old and new approaches must be discussed. Is it the "old" extrapolation method that the authors call the "ROPP extrapolation"? Or does ROPP use a different method? What is the reason of the failure of the old extrapolation technique for the GNOS data?

**Author's response:** We agree that the "standard" ROPP should be described in more detail in the revised manuscript.

In the context of the difficulties processing GNOS data, ROPP includes a pre-processing step in order to correct degraded L2 data. The approach is based on Gorbunov et al (2005,2006), and it is used routinely for other GPS-RO missions. Briefly, smoothed L1 and L2 bending angle and impact parameters are computed. An impact height, PC, above which the L2 data is considered reliable, is estimated using an empirical "badness score". The empirical badness score at time $t$, is

defined as,

$$Q(t) = \left( \frac{abs(\overline{p_1(t)} - \overline{p_2(t)})}{\Delta p_a} + \frac{\delta p_2(t)}{\Delta p_b} \right)^2$$

where $\delta p_2$ is a measure of the width of the L2 spectrum, $\overline{p_1(t)}$ and $\overline{p_2(t)}$ are the L1 and L2 impact parameters, respectively, computed from smoothed timeseries, $\Delta p_a$=200 m and $\Delta p_b$=150 m (See also, Eq. 11 Gorbunov et al, 2006 for a slightly modified form). The largest $Q(t)$ value in the impact height interval between 15 km to 50 km is stored as the badness score for the occultation, potentially for quality control purposes.

The mean L1 and L2 bending angle and impact parameters are then computed in a 2 km impact parameter interval directly above PC. Simulated L2 bending angles and impact parameters are computed by adding the mean (L2-L1) differences to both the L1 bending angle and impact parameter values, using the data in the 2 km interval. Simulated L2 and L1 phase values are then computed from these bending angles. Corrected L2 excess phase values are computed by merging the observed L2 phase above PC, with the simulated values below PC, using a smooth transition over 2 km, centred on PC. The corrected L2 phase values are subsequently used in the wave optics processing of the L2 signals.

A difficulty with the GNOS processing is related to determining the impact height PC, used for both the computation of the mean L1 and L2 differences, and defining the transition between observed and modelled L2 phase values. Although the "badness score" is used to determine PC, PC also has a maximum value (20 km). This is defined as the wave optics processing height (25 km) minus a 5 km "safety border". Therefore, the mean bending angles and impact parameters used in the L2-L1 correction can only be computed in a 2 km interval up to a maximum impact height of 22 km. Unfortunately, this is not high enough for GNOS L2 signals, with the result that the mean L2-L1 bending angle and impact parameters computed in the 2 km interval above PC are corrupted.

M. E. Gorbunov, K. B. Lauritsen, A. Rodin, M. Tomassini, and L. Kornblueh (2005), Analysis of the CHAMP Experimental Data on Radio-Occultation Sounding of the Earth's Atmosphere, Izvestiya, Atmospheric and Oceanic Physics, 41, No. 6, 726–740.

Gorbunov, M. E., K. B. Lauritsen, A. Rhodin, M. Tomassini, and L. Kornblueh (2006), Radio holographic filtering, error estimation, and quality control of radio occultation data, J. Geophys. Res., 111, D10105, doi:10.1029/2005JD006427.

**Changes in the manuscript:** We add the description for the failure of ROPP software processing GNOS observations in the section 1 and 2, at p4 line 1 to 20, p6 to p7 in the track changes version. 4.

**Reviewer's comment:** The paper by Zou and Zeng is in the reference list, but is not discussed nor referenced in the text. Please provide a comparative analysis of the old and new QC methods with the explanation of why the old QC methods are not sufficient for your data analysis. In particular, will the "badness score" introduced by Gorbunov et al. and successfully applied for CHAMP, COSMIC, METOP and other observations, be also useful for the FY3C/GNOS data analysis? If not, why?

**Author's response:** Thanks for the comments. More references will be cited and discussed in the revised manuscript. Originally, we'd like to find out a method to identify the quality of GNOS

profiles based on physical meaning and without using background data, just as the "badness score". When we look at the performance of "badness score", it is not suitable for GNOS (see fig1). The values of L2 badness score range from 15 to 1000 plus. The reason might be related to some empirical parameters. The explanation can be partly found in the previous answer. Other missions work well using "badness score" since the lowest straight line tangent altitude of L2 is low enough. When discussed with scientists from EUMETSAT, GRAS can get down to 15km for more than 90%. But it is not the case for GNOS. Only 70% of L2 signal can be reached below 20km. So the noise_estimate parameter as the quantity evaluation of the new L2 extrapolation method is used as a quality indicator, which could show the performance of L2 extrapolation and identify the bad profiles.

[Figure]

Fig 1. The cases of L2 badness score fail our QC and pass our QC

**Changes in the manuscript:** please see the p10, from Line 15 to 28 in the track changes version.

5.

**Reviewer's comment:** Page 9 (lines 3–7): The physical meaning of noise_estimate is easy to understand.

What is easy to understand is the fact that $\Delta\alpha$ is restricted to be close enough to its estimate obtained from a simple ionospheric model. Nevertheless, it is a good idea for the authors to explicitly mention this rather than appeal that something is "easy to understand". Still, some questions remain. Does n in formula (4.1) stay for refractivity of number of data? Number of data is definitely missing somewhere, because the sum in this formula needs to be normalized by the number of data. If n is refractivity, at what height is it taken? Provide explanations or definition regarding n.

**Author's response:** Thanks for the suggestion. n in formula 4.1 is the number of data. This will be fixed in the revised manuscript.

**Changes in the manuscript:** please see the p11, line 11 to 22 in the track changes version.

6.

**Reviewer's comment:** Page 11 (lines 21–22): The GRAS standard deviations are worse in the troposphere might due to sampling; essentially GRAS is able to measure more difficult cases. This statement needs more explanation. What are "more difficult cases"? Do they mostly occur in tropics? Can the authors provide any examples? Is it possible to evaluate a regionalized statistics (tropics, mid-, and polar latitudes)?

**Author's response:** The comparison between GRAS and GNOS is not the most important part of

the manuscript, thus a general remark is made. Regionalized statistics results can be seen in SC5 by Sean Healy. For more details of different kinds of statistics can be found on ROM SAF web pages. GNOS occultations are routinely available from the ROM SAF web pages. See, http://www.romsaf.org/monitoring/matched.php

**Changes in the manuscript:** Please see the p14 from line 9 to 22 in the track changes version.

[revised manuscript text omitted]

**For reviewer #3**

**Responses to the specific comments**

1.

**Reviewer's comment:** Introduction. The manuscript lacks motivation. Since the authors present a new methodology to correct the L2 signal bending, the "old" ROPP L2 signal correction approach should be described. Additionally, the differences between the "old" and the "new" approaches should be emphasized and discussed in detail. Currently, the reader cannot understand why the current ROPP approach does not work for the GNOS retrievals and all relevant references are missing.

**Author's response:** Thank you for pointing out the problem. We will add the relevant references and additional discussion about the old and approaches to clarify the GNOS retrievals in the revised manuscript. In the response to the reviewer #2, we explain why the current ROPP approach does not work for the GNOS. Generally, the old approach requires the L2 penetrating down into 20km at least.

**Changes in the manuscript:** In the track version manuscript for review #3, the introduction is reworded in P4 from line 2 to 17. The reason for the failure of ROPP processing for GNOS is described in detail in section 2, that's from P5 to P7.

2.

**Reviewer's comment:** Introduction: P. 3; Line 30. "These biases are not seen with other RO missions." Yes, the L2 signal is weaker than the L1 signal. However, other RO missions do not lose L2 signal tracking that much high up in the neutral atmosphere. The authors should explain why GNOS loses L2 signal tracking in the stratosphere at ~ 20 km, unlike all other RO missions. The authors state that the most prominent quality issue was the large departures biases, in the vertical range of 5 − 30 km. This altitude covers the middle troposphere up to the middle stratosphere. Then, within this context, if GNOS loses 30% of the profiles below 20 km (see P. 5; Line 11), then the authors should explain how does GNOS contribute to Numerical Weather Prediction (NWP) and specify the most effective altitude range of the GNOS RO profiles.

**Author's response:** The reason for GNOS losing L2 signal tracking is that GNOS has a lower SNR compared to other missions. Additionally, the GNOS antenna is smaller and not well located on the satellite. Consequently, we have to use additional cables, which results in a larger decrease of SNR than expected. Scientists from EUMETSAT confirmed that GRAS can get down to 15 km for more than 90% of the cases, but it is not the case for GNOS. Only 70% of L2 can reach below 20km. However, note that GNOS on FY3C is just the first Chinese GPS-RO mission. For the second satellite, FY3D, GNOS has more antenna units and in turn, has higher SNR than FY3C. Thus, the L2 signal tracking gets better. The proportion of the large departures biases in FY3D is smaller than in FY3C as well.

[Figure]

[Figure]

[Figure]

[Figure]

Figure 1

It is true that GNOS initially lost 30% of the profiles below 20 km, but that was before applying the new L2 extrapolation method outlined in the paper. After adopting the new method, we can process more GNOS profiles successfully. .

Regarding the impact on numerical weather prediction, GNOS was tested in the ECMWF assimilation system for the period November 23, 2017 to March 5,2018, prior to operational assimilation in the ECMWF system in March 2018. GNOS is assimilated operationally in the impact height interval from 8 km to 50 km in the extra-tropics, and from 10 km to 50 km in the

tropics. Although the medium-range forecast scores were generally neutral, in the short-range, the assimilation of GNOS data clearly improved the fit to other GPS-RO data, such as Metop GRAS A,B GRAS, COSMIC-6 etc. Figure 2 shows the improvement in the GPS-RO departure statistics for short-range forecasts when GNOS data is assimilated. This Figure could be added the final manuscript, but the main focus of the paper is how the current operational FY3C GNOS data is processed, rather than the impact in NWP systems.

[Figure]

Figure 2: The percentage change in the GPS-RO departure statistics as a result of assimilating the GNOS measurements. The change in the standard deviation of the background (o-b) departures are on the right, and the analysis (o-a) departures are on the left. The statistics are globally averaged, and the dotted lines indicated 95 % statistical significance. Values less than 100 % on the left hand side indicate that the short-range forecasts fit the other GPS-RO data more closely as a result of assimilating GNOS.

**Changes in the manuscript:** we add the followings in the p4, line3. "The reason for GNOS losing L2 signal tracking is that GNOS has a lower SNR compared to other missions. Additionally, the GNOS antenna is smaller and not well located on the satellite. Consequently, we have to use additional cables, which results in a larger decrease of SNR than expected."

To reduce misunderstanding, the sentence regarding the assimilation progress is removed to the last paragraph of the manuscript. P4, line 22 to 25.

3.

**Reviewer's comment:** New L2 extrapolation: Equation (3.4) states that the bending angle in L2 frequency equals the bending angle in L1 frequency plus a correction factor, which is proportional to the ionospheric TEC. The problem in Equation (3.3) is that it is derived using Equation (3.2), which is valid only for ionospheric bending and not for neutral atmosphere bending, as specifically mentioned in Culverwell and Healy (2015). Within the neutral atmosphere the

ionospheric bending becomes negligible and the signal bending at tropospheric and stratospheric altitudes has an exponential dependency on the impact parameter – different than Equation (3.2). Therefore, how could the authors apply Equation (3.2) to correct for the L2 bending angle within the neutral atmosphere using bending angle approximations derived for ionospheric bending only – particularly when applying this method from the lowest altitude the L2 signal is lost and 20 km up with a maximum upper limit of 70 km that is around the bottom side of the ionospheric D layer?

**Author's response:** As to this comment, Sean Healy gave a detailed response in SC1.
**Changes in the manuscript:** This part is reworded in P8 to P9.
4.
**Reviewer's comment:** Equation (4.1): The Xso is estimated from the least squares fit between the observed L1 and L2 bending angles. Then again, the new noise_estimate the authors introduce defines a new statistical metric based on how close the Xso is to the observed L1 and L2 bending angle difference. But, the Xso was estimated in Equation (3.4) to fit the minimum bending angle difference in L1 and L2. This noise_estimate appears to be misleading, without physical underpinning and with an over-fitting nature that beats down the scatter. Additionally, P. 9; Line 8: "The physical meaning of noise_estimate is easy to understand." Is not easy to understand and the authors should explain the rationale of defining it, because the Xso has already been estimated well via Equation (3.4). Also, how do the authors decide on the 20 microradiances as the threshold value?

**Author's response:** The "noise_estimate" provides information about how well we are able to fit the L2-L1 bending angle differences in the in the fitting interval where we trust the data, using the retrieved value Xso. Hence, the noise estimate is the least squares solution cost function value, divided by the number of points in the 20 km fitting interval. The fitting model is physically based, albeit assuming a simple ionospheric model, as discussed below. If the fitting model can reproduce the L2-L1 bending angle differences accurately, we can use the Xso to extrapolate the L2-L1 differences below 25 km, to produce ionospheric corrected bending angles used for NWP applications. The 20 microradian threshold is empirical, but it is informed by the assumed bending angle error statistics used in the assimilation of GNSS-RO data. Typically, the assumed bending angle error is 1.25 % from around 10 km to ~32 km. For example, this translates into around 6 microradians at 25 km, increasing to 13 microradians at 20 km. The 20 microradian threshold is designed to screen out cases where the L2-L1 extrapolation could introduce significant additional errors. We agree that the "easy to understand" statement should be clarified and expanded upon. However, the "over-fitting" comment is not clear.
**Changes in the manuscript:** The changes can be found in P10, line 4 to 19.
5.
**Reviewer's comment:** Section 4.2: The authors do not explain why is it necessary to monitor the performance of GNOS mean L1 and L2 phase delays in the height interval of 60 to 80 km. Also, why the mean phase and not the phase variation with altitude within this height range? What GNOS product is assimilated in NWP models and how does monitoring the 60-80 km phase delays help us to QC the profile below?

**Author's response:** We take these phase delays as one of QC factors because empirically it was found to determine the performance of GNOS when compared with reanalysis data. When encountering the bad profiles, the rising L1 and L2 mean phase delays have small values. The result is only based on FY3C. Subsequently, when we look at FY3D, this phenomenon disappears. Thus this factor is not a general one. We are considering cutting this part of from the manuscript.

**Changes in the manuscript:** Although the mean phase delay is not suit for FY3D, it is still kept as it is applied in FY3C. The phase variation with altitude is also a good way to monitor the performances of the observations. Beyerle et al. (2004) also suggested a QC approach to reject the RO observations degraded by ionospheric disturbances based on the phase delay of L1 and L2 signals. GNOS bending angle profiles are assimilated in NWP. Excess phase is the input of bending angle inversion. It's better to monitor the near-raw observations before messing up with the following processing. The correction is made at P14 in the track changes version.

6.

**Reviewer's comment:** P. 10; Line 21: "...these have been tested with one day of data..." The statistical sampling used in the determination of the statistical performance of the QC methods is low and does not represent the statistical performance of the GNOS profiles around the globe and under different seasons.

**Author's response:** One day of data was used to initially estimate the various QC parameters and then these were tested over longer periods. Clearly, the new L2 extrapolation method is rather effective at eliminating the large errors for the longer period, globally (See Figure 13,14)The plot shown here is just an example.

**Changes in the manuscript:** We carried out a new statistics using three months of data April 1 to June 30, 2018. In the corresponding part, we reinterpret the performance of the QC. Please see the detailed changes in the track version in P15 line 14 to 26.

7.

**Reviewer's comment:** Section 5: The authors explanation of the 15% disagreement between the GNOS and GRAS profiles below 10 km is inadequate. Ideally, collocated profiles between GNOS and GRAS should be used to quantify the degree of agreement or disagreement. However, if there are not enough collocated profiles between July 6 and August 2, 2018, perhaps the authors could use the entire time period GNOS provides RO profiles and if there are still not enough collocated profiles the authors could bin their profiles either into latitude sectors or seasons and then compare with GRAS to create an ensemble study to greatly increase the statistical sampling. The results represent a limited statistical sampling to support the authors' claims.

**Author's response:** Statistics for matched occultations are routinely available from the ROM SAF web pages. See,

http://www.romsaf.org/monitoring/matched.php

An example for GNOS versus Metop-A GRAS is attached. The GNOS data presented on these pages is processed with the method outlined in the paper. However, we do not believe that the matched occultation statistics provide any additional information, relative to the bending angle departure statistics computed with an accurate short-range forecast.

**Changes in the manuscript:** We reword this part. Please see P17 from line 4 to 17.

Minor comments

a)

**Reviewer's comment:** P. 2; Line 16: "...velocity and anti-velocity antennas..." Do you mean fore and aft antennas?

**Author's response:** Yes

b)

**Reviewer's comment:** P. 2; Line 19: What is the GNOS inclination in Table 1?

**Author's response:** The inclination of FY3C/GNOS is 98.75 ˚

**Changes in the manuscript:** We add the inclination of GNOS in Table 1.

c)

**Reviewer's comment:**P. 2; Line 17: Is BDS global or region constellation. Mention geographic restrictions of RO.

**Author's response:** BDS both has global and region constellation. The distribution of BDS RO can be shown as follows, also it can be refered to Mi Liao et al.,2016

[Figure]

Figure 3. Map of the GNOS BDS occultation coverage from
1 November to 31 December 2013, with a total of 4648 samples.
Different colours indicate different penetration depths.

[Figure]

Figure 4. Map of the GNOS BDS occultation coverage

Different colours indicate different constellations. MEO have the same altitude as GPS.

d)

**Reviewer's comment:**P. 3; Line 22: "...departure statistics..." From what?

**Author's response:** From background data, such as forecast data.

**Changes in the manuscript:** This part is overlapped with the first part of the section 2. For better elaboration, we decide to delete this part in the revised manuscript. The correction is at p3 line 22-30 in the track changes version.

e)

**Reviewer's comment:** P. 3; Line 25: Why more than 20% levels of the profile? How was this threshold selected? Explain.

**Author's response:** Compared with background data, the bad profiles are defined as the mean biases greater than 10% (100*(O-B)/B) from 5km to 30 km. As we know that the bias of RO at that height is about 1% in normal case. If the threshold is set as 10%, the large departure profiles can be identified.

**Changes in the manuscript:** This part is overlapped with the first part of the section 2. For better elaboration, we decide to delete this part in the revised manuscript. The correction is at p3 line 22-30 in the track changes version.

f)

**Reviewer's comment:** P. 4; Line 10: What is the most effective altitude range that GNOS provides the best RO profiles and explain how this information is used in NWP and how does it improve NWP. Include references to support claims.

**Author's response:** Currently, there are no published papers talking about the GNOS in NWP. Only some technical reports from personal communications. However, see Figure 2 above.

**Changes in the manuscript:** The impact of NWP using GNOS is not the main focus of the paper.To reduce misunderstanding, the sentence regarding the assimilation progress is removed to the last paragraph of the manuscript just as a general remark. P4, line 22 to 25.

g)

**Reviewer's comment:** P. 4; Line 14: "...may..." replace with "...could be..."

**Author's response:**   Fine.

**Changes in the manuscript: Done.**

h)

**Reviewer's comment:** P. 5; Line 11: Is this L2 signal loss at 20 km normal? Usually L2 signal is lost in the middle troposphere which is about 5 km. Explain.

**Author's response:**   This can be seen from my reply to your second major comment.

**Changes in the manuscript:** We add the explanation in P4 line 14 to 17 of the track changes version.

i)

**Reviewer's comment:** P. 5; Line 27: "...consistency..." replace with "...agreement..."

**Author's response:**   Fine.

**Changes in the manuscript:** Done.

j)

**Reviewer's comment:** P. 6; Line 5: Define "obvious errors".

**Author's response:**   Fine.

**Changes in the manuscript:** Reword as "large bending angle and refractivity departures" in P.8 line12 of the track changes version.

k)

**Reviewer's comment:** P. 6; Line 9: Define "other profiles".

**Author's response:**   Fine.

**Changes in the manuscript:** Delete the ambiguous words.

l)

**Reviewer's comment:**P. 6; Line 11: This definition of the ionosphere is crude, general, and unrealistic. Usually, the ionosphere is represented with multiple Chapman profiles with different scale heights. Mathematically, the Dirac function obtains a value of 0 at altitudes outside a very small neighborhood of the peak height.

**Author's response:**   The ionospheric model is crude, and it would not be valid if we were attempting to retrieve ionospheric information. However, we are only interested in modelling the impact of the ionosphere on bending angles with a tangent height well below the ionosphere, typically in the 25-60 km vertical interval. The ionospheric bending in this interval varies slowly with height (impact parameter). For example, adding a sporadic E layer would not change the shape of the L2-L2 difference curve below 60 km significantly. Conversely, we cannot retrieve an E-Layer from the L2-L1 differences below 60 km. Some authors assume that the L2-L1 is a constant. We use the delta function model because it produces a more realistic, slow variation of L2-L1 with height.

**Changes in the manuscript:** Please see the changes made in the P8 line 27 to P9 line7, and P10 line 4 to 9.

m)

**Reviewer's comment:** P. 6; Line 27: Why the peak height is 300 km? What led to this selection? The rule of thumbs says that per 100 km different in ionospheric shell height leads to 1 TECU error in the ionospheric total electron content. How sensitive is the estimation of Xso to the ionospheric TEC?

**Author's response:** Xso should be proportional to the ionospheric TEC because the L2-L1 differences should be proportional to the TEC. However, we are not trying to retrieve the TEC here. We estimate Xso in order to extrapolate the L2-L1 differences below 25 km using a reasonable curve. We apply the Chapman layer ionospheric model. Statistically, the peak height is around 300km, see the Culverwell and Healy, 2015 (ROM SAF). Experiments for testing the sensitivity of the peak height from 250km to 350km, in 10km increments, show that the final corrected bending angle is not sensitive to the peak height. The largest difference is about $10^{-5}$ urad. The plot (not shown here) is hard to differentiate the different results. Thus we think the 300km is reasonable. This will be noted in the revised paper.
**Changes in the manuscript:** Please see the changes made in the P10 line 11 to 19.

n)
**Reviewer's comment:** P. 7; Equation (3.4): This equation describes the ionospheric bending angle and not the neutral atmosphere. How can the authors apply this equation to correct for the L2 bending in the neutral atmosphere?

**Author's response:** This can be found in the comment of SC1 by Sean Healy.
**Changes in the manuscript:** This can be found the explanation in the Part3 of the track changes version.

[revised manuscript text omitted]